# Navigation with QPHIL: Quantizing Planner for Hierarchical Implicit Q-Learning

## Abstract

Offline Reinforcement Learning (RL) has emerged as a powerful alternative to imitation learning for behavior modeling in various domains, particularly in complex navigation tasks. An existing challenge with Offline RL is the signal-to-noise ratio, i.e. how to mitigate incorrect policy updates due to errors in value estimates. Towards this, multiple works have demonstrated the advantage of hierarchical offline RL methods, which decouples high-level path planning from low-level path following. In this work, we present a novel hierarchical transformer-based approach leveraging a learned quantizer of the space. This quantization enables the training of a simpler zone-conditioned low-level policy and simplifies planning, which is reduced to discrete autoregressive prediction. Among other benefits, zone-level reasoning in planning enables explicit trajectory stitching rather than implicit stitching based on noisy value function estimates. By combining this transformer-based planner with recent advancements in offline RL, our proposed approach achieves state-of-the-art results in complex long-distance navigation environments.

## 1 Introduction

Navigation and locomotion in complex, embodied environments is a long-standing challenge within Machine Learning (Kaelbling et al., 1996; Sutton & Barto, 2018). Operating non-trivial agents in complex spaces is critical in a wide range of real-world applications, such as in robotics or in the video game industry. A core difficulty of navigation lies in solving long-horizon tasks that require intricate path planning (Hoang et al., 2021; Park et al., 2024). In the Reinforcement Learning (RL) setting (Sutton & Barto, 2018), traditional online Goal-Conditioned deep Reinforcement Learning (GCRL) methods often struggle with such long-horizon tasks because of the sparse nature of the reward signal, leading to hard exploration problems. Offline GCRL circumvents this exploration problem by leveraging large amounts of unlabeled and diverse demonstration data to learn policies through passive learning (Prudencio et al., 2023). A core advantage of offline RL compared to other forms of behavior extraction from datasets – for instance imitation learning – is the ability to improve over suboptimal datasets, e.g. by learning state value functions to bias the learned policy towards rewarding actions (Kostrikov et al., 2021). However, offline RL is not trivial to apply for long-horizon goal-reaching tasks, which often provide sparse reward signals, leading to a noisy value function which consequently hinders the performance of the policy. Part of this issue comes from low "signal-to-noise" ratio to learn the value function (Park et al., 2024). Because a suboptimal action can be corrected quickly in subsequent steps of a trajectory, its impact on the real value for faraway goals can be covered by the value prediction noise, which can lead to the learning of suboptimal behaviors for the policy.

Hierarchical architectures have shown considerable advantages in goal-conditioned navigation to solve such issues (Vezhnevets et al., 2017; Pertsch et al., 2021; Park et al., 2024). These approaches effectively decompose the problem into two distinct components: high-level path planning and low-level path following. Among these, Hierarchical Implicit Q-Learning (HIQL) (Park et al., 2024) has recently emerged as the state-of-the-art method. HIQL leverages a hierarchical structure to learn both a high-level policy for generating subgoals and a low-level policy for achieving these subgoals, all within an offline reinforcement learning framework.

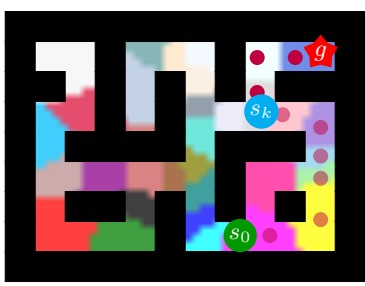

Figure 1: QPHIL relies on learning discrete *landmarks* (zones) to reduce navigation to high-level landmark sequence generation and low-level landmark-conditioned path-following.

Although introducing a hierarchical structure enhances performance by improving the signal-to-noise ratio at each level, it only partially alleviates the issue. For long-distance tasks, the signal-to-noise ratio still degrades during subgoal generation, which can result in a noisy high-level policy and, consequently, reduced performance. In this paper, we propose to shift the learning paradigm of the high-policy towards discrete space planning. Towards this, we first train a Vector Quantized Variational Autoencoder (VQ-VAE) (Van Den Oord et al., 2017) on the state-space from which we extract its quantized representation, namely leading to a clustering of the state space into what we propose to refer to as *landmarks*. We ensure the temporal consistency of the obtained landmarks through a contrastive regularization of the VQ-VAE loss. Then, we leverage a transformer architecture (Vaswani et al., 2017) to extract a discrete high-level policy, enabling consistent landmark-level planning in the discrete space representation (see Figure 1). Finally, we train a combination of a low-level landmark-conditioned policy and a low-level goal conditioned policy to solve the subgoal and last-goal navigation tasks respectively.

Our main contribution is to introduce **Quantizing Planner for Hierarchical Implicit Learning (QPHIL)**, a novel approach that addresses long-range navigation by combining the strengths of VQ-VAE, transformers, and offline reinforcement learning. We first evaluate QPHIL on the established AntMaze navigation tasks Fu et al. (2020), then introduce a novel, more challenging variant, *AntMaze-Extreme*, along with two associated datasets, better suited to study long-distance navigation. In both settings, we demonstrate that QPHIL significantly outperforms prior offline goal-conditioned RL methods, particularly in large-scale settings.

## 2 RELATED WORK

**VQ-VAEs for reinforcement learning** Vector Quantized Variational Autoencoders (Van Den Oord et al., 2017) have demonstrated their utility in reinforcement learning., particularly for offline setups. Jiang et al. (2022) and Luo et al. (2023) use a VQ-VAE to discretize continuous action spaces, mitigating the curse of dimensionality. In contrast, QPHIL uses a VQ-VAE to quantize states, simplifying waypoint generation in long-horizon tasks. These two approaches are complementary: action quantization focuses on simplifying policy search, while state quantization helps in task decomposition such as in Hamed et al. (2024) where a VQ-VAE is used to define landmarks as states like in QPHIL but without a contrastive loss and in the online setting. Within goal-conditioned reinforcement learning, several works used VQ-VAE to encode observations into discrete (sub)goals. Lee et al. (2024a) learn quantized goals to simplify curriculum learning. Islam et al. (2022) also use quantization to map (sub)goals into discrete and factorized representations to efficiently handle novel goals at test time. Both these works focus on online goal-conditioned reinforcement learning while QPHIL tackles the offline learning setting. Kujanpää et al. (2023) also uses VQ-VAE to generate discrete subgoals, but while they rely on a finite subgoal set derived from offline learning, QPHIL identifies subgoals directly through the VQ-VAE. To the best of our knowledge, VQ-VAE has never been applied for discrete planning in offline GCRL settings.

**Hierarchical offline learning** Leveraging a hierarchical structure to decompose and simplify sequential decision-making problems is a long-standing idea and subject of research within machine learning (Schmidhuber, 1991; Sutton et al., 1999). Recently, multiple successful works renewed the interest of the research community to these models, both for online (Vezhnevets et al., 2017; Pertsch et al., 2021; Kim et al., 2021; Fang et al., 2022) and offline reinforcement learning (Nachum et al., 2018b; Ajay et al., 2020; Rao et al., 2021; Rosete-Beas et al., 2023; Yang et al., 2023; Shin & Kim, 2023). Among these, Hierarchical Implicit Q-Learning (HIQL) from Park et al. (2024) has emerged as the state-of-the-art method for offline goal-conditioned RL. Most methods differ in how they represent and use subgoals; for instance, Nachum et al. (2018a) compare different subgoal representations, while Ajay et al. (2020) focus on detecting and combining primitive behaviors. While

all aforementioned works focus on continuous state representations, QPHIL leverages the advantage of discrete state representations to simplify long-distance navigation.

**Planning** Hierarchical planning methods are a natural complement to RL. For instance, HIPS (Kujanpää et al., 2023) learns to segment trajectories, generating subgoals of varying lengths to facilitate adaptive planning. HIPS-$\varepsilon$ (Jiang et al., 2023; Kujanpää et al., 2024) introduces hybrid hierarchical methods and offers completeness guarantees, but their method is only usable with a discrete state space. Li et al. (2022) proposes a method combining a high-level planner, based on a VAE, with a low-level offline RL policy. The VAE serves as the planner by leveraging the low-level policy's value function. QPHIL, on the other hand, utilizes a simpler subgoal planning approach directly tied to a discrete state-space via VQ-VAE.

**Transformers for (hierarchical) offline RL** Transformers have recently found application in hierarchical setups to solve RL problems. In Correia & Alexandre (2023), transformers are used for hierarchical subgoal sampling, where a high-level transformer generates subgoals for a low-level transformer responsible for action selection. Similarly, Badrinath et al. (2024) combine a Decision Transformer (DT) from Chen et al. (2021) with a waypoint-generation network. Ma et al. (2024) present a hierarchical transformer-based approach outperforming both aforementioned methods. Li et al. (2022); Ma et al. (2024) are extensions of DT that allows stitching, a known issue with Decision Transformer (Fujimoto & Gu, 2021; Emmons et al., 2021; Kostrikov et al., 2021; Yamagata et al., 2023; Xiao et al., 2023). Their approach, Goal-prompted Autotuned Decision Transformer (G-ADT) (Ma et al., 2024), is able to efficiently integrate an offline RL learning scheme similar to HIQL. G-ADT uses a low-level transformer and a simple high level subgoal policy, while we focus on the opposite scenario: our transformer is used to generate a plan of subgoal, which are then followed by a fully connected neural network. G-ADT uses an offline reinforcement learning objective to train its high-level policy while we rely on a simpler imitation learning objective.

**Sequence generation** Lee et al. (2024b) explore sequence generation using tokens learned via a VQ-VAE, much like QPHIL; but their work does not extend to reinforcement learning. In RL, sequence models have been used for various components (Bakker, 2001; Heess et al., 2015; Chiappa et al., 2017; Parisotto et al., 2020; Kumar et al., 2020). However, none of these approaches integrate planning over a discrete representation, which is central to our method. Sequence generation also plays a key role in offline RL, where it is helping to prevent out-of-distribution actions (Fujimoto et al., 2019; Kumar et al., 2019; Ghasemipour et al., 2021). Trajectory Transformer (Janner et al., 2021) completely treats offline RL as a sequence generation problem. They use a transformer to perform planning using Beam-Search and an IQL-like value function. Contrary to QPHIL, they use a simple dimension-wise uniform or quantile discretization.

**Spatial zone learning and skill discovery** Zone-based spatial navigation and skill discovery are additional techniques that overlap with hierarchical RL and QPHIL. For instance in the context of online RL, Kamienny et al. (2022) proposes to discover a set of "easy-to-learn" short policies, each corresponding to a given skill defined as an area of the state space, which can be eventually composed for the task at hand. Using a VQ-VAE, Mazzaglia et al. (2022) define skills that are used to influence exploration in online policies. In a closer context with a known state space, Gao et al. (2023) segments space into zones and uses these segments to facilitate goal-reaching tasks, but their method is restricted to vision environments. Similarly, Hausman et al. (2017) uses a multi-modal imitation learning approach to discover and segment skills from unstructured demonstrations, which is close to the discrete landmark generation in QPHIL. In skill decision transformer Sudhakaran & Risi (2023), a transformer is used to predict actions from latent variables discovered by a VQ-VAE. While their approach is close to QPHIL, it differs by not using contrastive loss and not planning subgoals.

## 3 PRELIMINARIES

**Offline Goal Conditioned Reinforcement Learning** We frame our work in the context of offline goal-conditioned Reinforcement Learning (offline GCRL), which involves training an agent to interact with an environment to reach specific goals, without getting access to the environment itself at train time. It is typically modeled as a Markov Decision Process (MDP) defined by a tuple

$(\mathcal{S}, \mathcal{A}, p, \mu, r)$, a dataset of demonstration trajectories $\mathcal{D}$ and a given goal space $\mathcal{G}$, where $\mathcal{S}$ is the state space, $\mathcal{A}$ the action space, $p(s_{t+1}|s_t, a_t) \in \mathcal{S} \times \mathcal{A} \to \mathcal{P}(\mathcal{S})$ the transition dynamics, $\mu \in \mathcal{P}(\mathcal{S})$ the initial state distribution and $r(s_t, g) \in \mathcal{S} \times \mathcal{G} \to \mathbb{R}$ the goal-conditioned reward function given the goal space $\mathcal{G}$. In GCRL, the reward is sparse as $r(s, g) = 1$ only when $s$ reaches $g$ and 0 otherwise. Hence, our objective is to leverage the dataset $\mathcal{D}$ composed of reward-free pre-recorded trajectoires $\tau = (s_0, a_0, ..., s_{T-1}, a_{T-1}, s_T)$ to learn the policy $\pi(a_t|s_t, g)$ to reach given goals $g \in \mathcal{G}$, by maximizing the expected cumulative reward, or return, $J(\pi)$, which can be expressed as: $J(\pi) = \mathbb{E}_{\substack{g \sim d_g \\ \tau \sim d_\pi}} \left[ \sum_{t=0}^{T} \gamma^t r(s_t, g) \right]$, where $\gamma$ is a discount factor, $d_g$ represents the goal distribution and $d_\pi$ is the trajectory distribution defined by: $d_\pi(\tau, g) = \mu(s_0) \prod_{t=0}^{T} \pi(a_t|s_t, g) p(s_{t+1}|s_t, a_t)$ and represents the trajectory distribution when the policy is used.

**Hierarchical Implicit Q-Learning (HIQL)** Because of the sparsity of the offline GCRL's reward signals, bad actions can be corrected by subsequent good actions, and good actions can be undermined by future bad actions in a trajectory. Hence, offline RL methods risk mislabeling bad actions as good ones, and vice versa. This leads to the learning of a noisy goal-conditioned value function: $\hat{V}(s_t, g) = V^*(s_t, g) + N(s_t, g)$ where $V^*$ corresponds to the optimal value function and $N$ corresponds to a noise. As the "signal-to-noise" ratio worsens for longer term goals, offline RL methods such as IQL Kostrikov et al. (2021) struggle when the scale of the GCRL problem increases, leading to noisy advantages estimates for which the noise overtakes the signal. To alleviate this issue, HIQL (Park et al., 2024) first proposes to learn a noisy goal conditioned action-free value function, inspired from IQL (Kostrikov et al., 2021):

$$\mathcal{L}_V(\theta_V) = \mathbb{E}_{(s_t, s_{t+1}, g) \sim \mathcal{D}} \left[ L_2^\tau(r(s_t, g) + \gamma V_{\hat{\theta}_V}(s_{t+1}, g) - V_{\theta_V}(s_t, g)) \right], \quad (1)$$

using an expectile loss $L_2^\tau(u) = |\tau - \mathbb{1}(u < 0)|u^2, \tau \in [0.5, 1)$ on the temporal difference of the value function, which aims at anticipating in-distribution sampling without querying the environment. This loss is then used to learn two policies with advantage weighted regression (AWR): $\pi^h(s_{t+k}|s_t, g)$ to generate subgoals on the path towards goal $g$ and $\pi^l(a_t|s_t, g')$, with $g' \in \mathcal{S}$ a given subgoal, to generate actions to reach the subgoals, in a hierarchical manner. With this division, each policy profits from higher signal-to-noise ratios, as the low-policy only queries the value function for nearby subgoals $V(s_{t+1}, s_{t+k})$ and the high policy queries the value function for more distant goal $V(s_{t+k}, g)$. More details about IQL and HIQL are availiable in the appendix A.

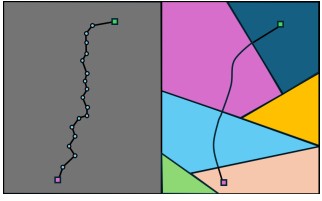
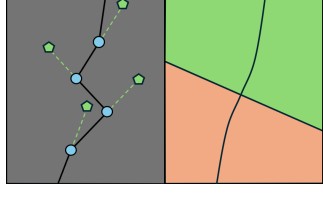
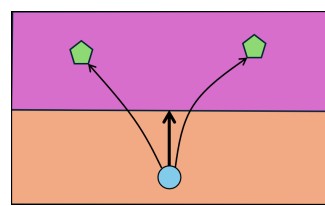

(a) Simpler high level planning     (b) Smoother low level targets     (c) Easier target conditioning

Figure 2: **Motivations behind QPHIL** (a) QPHIL aims to simplify the planning of the subgoals by leveraging discrete tokens. (b) By doing so, QPHIL avoids the noisy high-frequency target subgoal updates by updating the subgoal of the low-level policy after each landmark traversal only. (c) The subgoal reaching tasks are less demanding in conditioning for the low policy as it corresponds to the reaching of an entier subzone instead of a precise subgoal.

**Quantizing Planner for Hierarchial Implicit Learning (QPHIL)** If HIQL shows significant improvements in offline GCRL problems compared to previous flat-policy methods, its performance depends on the right choice of the subgoal step $k$. A high $k$ would improve the high policy's signal-to-noise ratio by querying more diverse subgoals but at the cost of decreasing the signal-to-noise ratio of the low policy. Conversely, a low $k$ would improve the low policy's signal-to-noise ratio by querying values for nearby goals but at the cost of the diversity of the high subgoals. Hence, HIQL might struggle for longer term goal reaching tasks as the low level performance imposes the choice of a sufficiently low $k$, which leads to high frequency noisy high subgoal targets for the low policy

to follow. QPHIL proposes to mitigate this issue by shifting the learning paradigm of the high policy into planning in a discretized learned space representation (see Figure 2).

# 4 QUANTIZING PLANNER FOR HIERARCHICAL IMPLICIT LEARNING

In this paper, we propose a new hierarchical goal conditionned offline RL algorithm: Quantizing Planner for Hierarchical Implicit Q-Learning (QPHIL). While current hierarchical methods rely heavily on continuous waypoint predictions, we propose to consider discrete subgoals, allowing to simplify the planning process. Instead of relying on precise coordinates, QPHIL identifies key landmarks to guide trajectory planning, much like how a road trip is described by cities or highways rather than specific geographic points. The algorithm detects these landmarks, creates landmark-based sequences, and formulates policies to navigate between them, ultimately reaching the target destination.

## 4.1 OVERALL DESIGN

QPHIL operates through four components: a state quantizer $\phi$, which divides the state set into a finite set of $k$ landmarks, a plan generator $\pi^{plan}$, which acts as a high-level policy to generate a sequence of landmarks to be reached given the final goal, and two low-level policy modules: $\pi^{\text{landmark}}$, which targets state areas defined as landmarks, and $\pi^{\text{goal}}$, which targets a specific state goal.

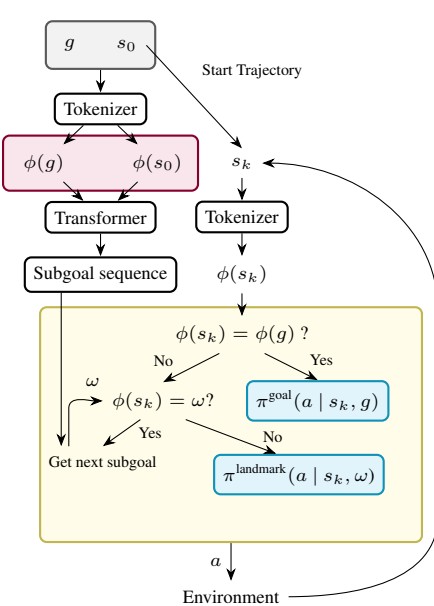

Figure 3: Inference pipeline of QPHIL (open-loop version, without replanning). Subgoal tokens are consumed from the sequence after each corresponding landmark is reached.

The state quantizer $\phi : \mathcal{S} \rightarrow \Omega$, with $\Omega \equiv [\![1, k]\!]$, is used to map raw states into a set of $k$ landmark indexes (or tokens), which serve as the building blocks for our planning strategy. These discrete representations are then processed by the plan generator, which assembles them into a coherent sequence that outlines the overall trajectory to be followed in the environment to reach the requested goal $g$. Given any history of tokens $(\omega_0, \cdots, \omega_s)_{s>0}$ and a targeted goal $g \in \mathcal{G}$, the plan generator produces sequences of tokens $(\omega_{s+1}, \cdots, \omega_{s+l})$ auto-regressively until $\omega_{s+l} = \dot{\omega}$ (with $\dot{\omega}$ an extra end of sequence token that stops the generation), following: $\pi^{\text{plan}} ((\omega_{s+1}, \cdots, \omega_{s+l}) \mid \phi(g), (\omega_i)_{i=0}^s) = \prod_{t=1}^l \pi^{\text{plan}} (\omega_{s+t} \mid \phi(g), (\omega_i)_{i=0}^{s+l-1})$. It is used from the start token $\omega_0 = \phi(s_0)$ of the episode to propose a sequence of landmarks to follow using $\pi^{\text{landmark}} : \mathcal{A} \times \mathcal{S} \times \Omega \rightarrow [0, 1]$, before navigating through the final goal $g$ using $\pi^{\text{goal}} : \mathcal{A} \times \mathcal{S} \times \mathcal{G} \rightarrow [0, 1]$. Optionally, it could also be used to re-plan after a given condition is reached (e.g. every $n$ steps or when the agent leaves its current area), which is mostly useful for complex environments with stochasticity.

Algorithm 1 (in appendix) presents the inference pseudo-code of our QPHIL approach. For a given goal $g$ and a start state $s_0$, the process first samples a new plan, which corresponds to a sequence of landmarks to be reached sequentially before navigating towards $g$. Then, until the first subgoal $\omega$ of this sequence is reached (which happens when $\phi(s) = \omega$), $\pi^{\text{landmark}}$ is used while conditioned on the current state and the sub-goal $\omega$ to sample actions to perform in the environment. When the sub-goal is reached, it is removed from the list, and we move to the next one, and so on until the list of sub-goals is empty. At that point, $\pi^{\text{goal}}$ is used to navigate towards the requested goal $g$. A graphical representation of the inference pipeline is available in Figure 3.

The remainder of this section presents our learning methodology for every component of our QPHIL approach. Each of the following components are learned sequentially.

## 4.2 Tokenization

As defined in the previous section, the state quantizer $\phi$ is used to map raw states into a set of $k$ tokens. It is composed of a state encoder $e : \mathcal{S} \to \mathbb{R}^d$, which maps the state space to a latent space of dimension $d$, and a learned code-book $z : \Omega \to \mathbb{R}^d$, which associates each token to a continuous embedding. From these, $\phi$ is defined as: $\phi(s) = \arg\min_{\omega \in \Omega} \Delta(e(s), z(\omega))$, with $\Delta$ the euclidean distance in the latent space.

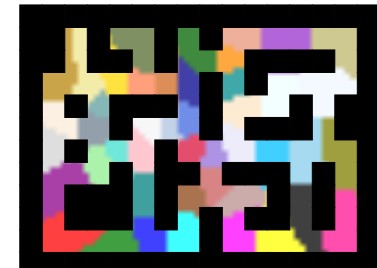

Figure 4: Tokenization example. Each token has an associated color. The color of each point in the background correspond to the color of the associated token. It is noticeable that the tokens align with the walls thanks to the contrastive loss.

As depicted in Figure 8, three losses are considered to train these components. First, the VQ-VAE learns a meaningful representation by considering a reconstruction loss, which ensures that states can be decoded from tokens they are projected into:

$$\mathcal{L}_{\text{recon}} = \mathbb{E}_{s \sim \mathcal{D}} \|f(z(\phi(s))) - s\|_2^2,$$

where $\mathcal{D}$ is the dataset and $f : \mathbb{R}^d \to \mathcal{S}$ is a decoder network trained together with the code-book of token embeddings.

As the gradient flow stops in the previous loss due to the discrete projection performed by $\phi$, we need to consider an additional commitment loss to learn the encoder parameters:

$$\mathcal{L}_{\text{commit}} = \mathbb{E}_{s \sim \mathcal{D}} \|e(s) - \text{sg}(z(\phi(s))\|_2^2 + \beta \|\text{sg}(e(s)) - z(\phi(s))\|_2^2,$$

where sg is the stop-gradient operator. The first term of this loss aims at attracting encoder outputs close to the code of the token states project into. As explained by authors of the VQ-VAE architecture (Van Den Oord et al., 2017), the embedding space grows arbitrarily if the token embedding does not train as fast as the encoder parameters. The second term, weighted by a hyperparameter $\beta$, aims to prevent this issue by forcing the encodings to commit to a code-book's embedding.

To introduce dynamics of the environment in the representation learned, we consider a third contrastive loss that incentivize temporally close states to be assigned the same tokens, while temporally distant states receive different tokens. To achieve this, we use the triplet margin loss (Balntas et al., 2016) applied to our setting:

$$\mathcal{L}_{\text{contrastive}} = \mathbb{E}_{\substack{s_t \sim \mathcal{D} \\ k \sim [\![-\delta, \delta]\!] \\ k' \sim \mathbb{Z} \setminus [\![-\delta, \delta]\!]}} \max\left\{\Delta(e(s_t), e(s_{t+k})) - \Delta(e(s_t), e(s_{t+k'})), 0\right\},$$

where $\delta$ is the time window used to specify temporal closeness of states in demonstration trajectories. This loss is of crucial importance in navigation, for instance in settings with thin walls, where two states can be close in the input space while corresponding to very different situations.

The tokenizer loss, used in training is then a linear combination of the three previous losses:

$$\mathcal{L}_{\text{tokenizer}} = \alpha_{\text{recon}} \mathcal{L}_{\text{recon}} + \alpha_{\text{commit}} \mathcal{L}_{\text{commit}} + \alpha_{\text{contrastive}} \mathcal{L}_{\text{contrastive}}.$$

## 4.3 Sequence generation

Once the tokenizer has been trained, each state from dataset trajectories can be discretized leading to sequences of tokens. Temporal consistency induces sub-sequences of repeated tokens corresponding to positions within a given region. By applying a simple post-processing step $\eta^\phi(\tau)$ that removes consecutive repetitions of tokens in a trajectory $\tau$, we obtain more concise sequences that succinctly represent key zones to traverse in the correct order. For instance, a tokenized sequence such as "1 1 1 2 2 3 3 3 4 4" is simplified to "1 2 3 4", reflecting the core structure of the trajectory. This process is illustrated in Figure 9.

Then, the planner $\pi^{\text{plan}}$ is trained following a teacher forcing approach on trajectories from $\mathcal{D}$, considering any future token of $\tau \in \mathcal{D}$ as the associated training goal:

$$\mathcal{L}_{\text{plan}} = -\mathbb{E}_{\tau \sim \mathcal{D}} \; \mathbb{E}_{\substack{t < |\tau|, \\ \phi(\tau_t) \neq \phi(\tau_{t+1})}} \; \mathbb{E}_{\substack{g \in \mathcal{G}, \\ \phi(g) \in \eta^\phi \tau_{>t}}} \; \log \pi^{\text{plan}}(\phi(\tau_{t+1}) \mid \phi(g), \eta^\phi(\tau_{\leq t})),$$

with $|\tau|$ the number of states in $\tau$, $\tau_t$ the $t$-th state in $\tau$, and $\tau_{\leq t}$ (resp. $\tau_{>t}$) the history (resp. future) of $\tau$ at step $t$. We implement $\pi^{\text{plan}}$ using a transformer architecture (Vaswani et al., 2017) $h_\theta$, where $\pi^{\text{plan}}(. \mid \phi(g), \eta^\phi(\tau_{\leq t}))$ is defined for any $\omega \in \Omega$ using a softmax on the outputs of $h_\theta(\phi(g), \eta^\phi(\tau_{\leq t}))$. While an alternative would be to consider a markov assumption stating that $\pi^{\text{plan}}(. \mid \phi(g), \eta^\phi(\tau_{\leq t})) \approx \pi^{\text{plan}}(. \mid \phi(g), \phi(s_t))$, we claim that dependency on the full history of the sequence is useful to anticipate the next token to be reached, as it can be leveraged to deduce the precise location of the agent in large landmark areas (which is unknown during plan generation).

We note that this behavioral cloning way for training the planner could be complemented by an offline RL finetuning, following a goal-conditionned IQL algorithm for instance. Rather, we propose to consider in our experiments a data augmentation process on top of our behavioral cloning approach, which allowed us to obtain similar results, with greatly lower computational cost. Our data augmentation process relies on the opportunity of accurate trajectory stiching that our quantized space offers. Assuming that it is easy for our low-level policy $\pi^{\text{landmark}}$ to reach, from any state $s \in \mathcal{S}$, any state $s'$ such that $\phi(s') = \phi(s)$, we propose to augment $\mathcal{D}$ with all possible mix of trajectories that pass through the same landmark and own a similar future quantized state. That is, $\forall (\tau, \tau') \in \mathcal{D}^2, t \in [\![0, |\tau|]\!], t' \in [\![0, |\tau'|]\!], l \in [\![t+1, |\tau|]\!], l' \in [\![t'+1, |\tau'|]\!]$, we have: $\phi(s_t) = \phi(s'_{t'}) \wedge \phi(s_l) = \phi(s'_{l'}) \implies ((s_i)_{i=0}^t, (s'_i)_{i=t'+1}^{l'}) \in \mathcal{D} \wedge ((s'_i)_{i=0}^{t'}, (s_i)_{i=t+1}^l) \in \mathcal{D}$. This technique is illustrated in Figure 10.

### 4.4 LOW-LEVEL POLICIES

As described above, two low-level policies $\pi^{\text{landmark}}$ and $\pi^{\text{goal}}$ have to be trained to perform actions in the environment. Both of these policies are trained via IQL, whose principle is described in section A. As $\pi^{\text{goal}}$ works with initial (non tokenized) goals to allow final precise navigation in the final area of the task, we consider a classical $V$ network $V^{\text{goal}}$ trained using equation 7 on raw states and goals, independently from all other components, from which the policy is extracted using AWR:

$$\mathcal{L}_{\pi^{\text{goal}}} = \mathbb{E}_{(s_t, s_{t+1}, g) \sim \mathcal{D}}[\exp(\beta \cdot (V^{\text{goal}}(s_{t+1}, g) - V_{\theta_V}(s_t, g))) \log \pi^{\text{goal}}(s_{t+1}|s_t, g)] \quad (2)$$

Our policy $\pi^{\text{landmark}}$ uses the same principle, but replaces the distant goal $g$ by sub-goals defined as the next landmark to be reached from quantized trajectories in the buffer of training transitions. That is, given any training trajectory $\tau = ((s_t, a_t, s_{t+1}, g))_{t=0}^{|\tau|-1}$, we consider a sequence of relabelled transitions $\tilde{\tau} = ((s_t, a_t, s_{t+1}, \text{next}(\tau, t))_{t=0}^{|\tau|-1}$, where $\text{next}(\tau, t) = \phi\left(s_{\min(\{l \in [\![t+1, |\tau|]\!] : \phi(s_l) \neq \phi(s_t)\})}\right)$ corresponds to the next different subgoal in the sequence of quantized states $(\phi(s_t))_{t=0}^{|\tau|-1}$ of the trajectory $\tau$. $V^{\text{landmark}}$ is then trained on that set of relabelled transitions $\tilde{\mathcal{D}}$. Then, the policy is extracted similarly as for $\pi^{\text{goal}}$ using equation 2 on $\tilde{\mathcal{D}}$ rather than $\mathcal{D}$, in order to obtain $\pi^{\text{landmark}}$ via a relabelled IQL from $V^{\text{landmark}}$.

## 5 EXPERIMENTS

Our experiments aim to address the following questions:

1. Does QPHIL architecture enable efficient long-term navigation?

2. Can QPHIL handle sparse data scenarios using token-level stitching?

3. Does QPHIL still performs in diverse state-target initialization?

4. What is the impact of the contrastive loss used for landmark learning?

Additional results on the impact of re-plannification or the use of RL at the higher policy level are given in appendix D.

### 5.1 EXPERIMENTAL SETUP

The following section provides details on environments, datasets and baselines used to study QPHIL.

**Antmaze** We measure QPHIL performance through a set of AntMaze environments of increasing sizes, providing challenging long-term navigation problems. In AntMaze the agent controls an 8-DoF ant-shaped robot. Observations consist in a 29-dimension state vector (e.g. positions, torso coordinates and velocity, angles between leg parts). For original AntMaze environments we use datasets provided in the D4RL library (Fu et al., 2020) as well as two additional datasets for even larger mazes, namely Antmaze-Ultra provided by Jiang et al. (2022) and Antmaze-Extreme which we created (Appendix C.2). For each maze type (medium, large, ultra and extreme), we train on two types of datasets: "play" and "diverse", each containing 1000 trajectories of 1000 steps. The "play" variant has been generated using hand-picked locations for the goal and starting positions while the "diverse" variant has been generated using random goal and starting positions.

**Baselines** We compare QPHIL (open-loop planning version) to 8 previous methods ranging from model-free behavior cloning and offline RL methods as well as model-based methods. For model-free methods, we include in our baselines the flat Goal-Conditioned Behavior Cloning (GCBC) (Ghosh et al., 2019) and the Hierarchical Goal-Conditioned Behavior Cloning (HGCBC) (Gupta et al., 2019). For offline RL, we include a goal-conditioned variant of Implicit Q-Learning (GC-IQL) (Kostrikov et al., 2021), a goal-conditioned variant of Policy-Guided Imitation (GC-POR) (Xu et al., 2022) as well as Hierarchical Implicit Q-Learning (HIQL) (Park et al., 2024). For model-based methods, we include Trajectory Transformer (TT) (Janner et al., 2021) which leverages a transformer architecture to model the entire flattened state-action sequence, Trajectory Autoencoding Planner (TAP) (Jiang et al., 2022) which performs model-based planning over discrete latent actions quantized by a VQ-VAE as well as Goal-prompted Autotuned Decision Transformer (G-ADT) (Ma et al., 2024), which generates sequences of subgoals in continuous space. We also include results from the very recent Planning Transformer (PT) method (Clinton & Lieck, 2024), which leverages the use of a decision transformer as an agent, that consumes a sequence of imagined next states of the agent (i.e., a plan) as input. As HIQL baselines and QPHIL ran on 8 seeded runs, results are provided with the mean ± std format.

## 5.2 DOES QPHIL ARCHITECTURE ENABLE EFFICIENT LONG-TERM NAVIGATION ?

Table 1: **Evaluating QPHIL on AntMaze environments.** We see that QPHIL scales well with the length of the navigation range, competing with SOTA performance on the smaller mazes and improving significantly on the SOTA on the larger settings.

| Dataset | GCBC | IQL | G-ADT | TAP | TT | PT | HGCBC | HIQL w/ repr. | HIQL w/o repr. | QPHIL w/ aug. | QPHIL w/o aug. |
|---|---|---|---|---|---|---|---|---|---|---|---|
| medium play | 71.9±16 | 70.9±11 | 82±1.7 | 78.0 | **93.3** | | 66.3±9.2 | 84.1±11 | 87.0±8.4 | 92.0±3.9 | 86.8±3.6 |
| medium diverse | 67.3±10 | 63.5±15 | 83.4±1.7 | 85.0 | **100** | 85.4±2.1 | 76.6±8.9 | 86.8±4.6 | 89.9±3.5 | 90.8±1.8 | 87.8±2.3 |
| large play | 23.1±3.5 | 56.5±14 | 71.0±1.3 | 74.0 | 66.7 | | 64.7±14 | 86.1±7.5 | 81.2±6.6 | 82.25±6.4 | **88.0±5.5** |
| large diverse | 20.2±9.1 | 50.7±19 | 65.4±4.9 | 82.0 | 60.0 | 82.3±5.8 | 63.9±10 | **88.2±5.3** | 87.3±3.7 | 80.25±3.3 | 80.5±7.4 |
| ultra play | 20.7±9.7 | 29.8±12 | | 22.0 | 20.0 | | 38.2±18 | 39.2±15 | 56.0±12 | **64.5±6.8** | 61.5±6.2 |
| ultra diverse | 14.4±9.7 | 21.6±15 | | 26.0 | 33.3 | 34.9±7.1 | 39.4±21 | 52.9±17 | 52.6±8.7 | 61.8±3.7 | **70.3±6.9** |

We first analyze the performance in success rate of the different baselines on the state-based AntMaze-{Medium,Large,Ultra} settings. As usual in D4RL Antmaze benchmarks (Park et al., 2024), starting state and target goals are sampled near two reference points, forcing the agent to walk across the entire maze. Figure 5 showcases examples of tokenizations obtained by our VQ-VAE model. One can observe a tendency for learned clusters to decrease in size in the middle of all mazes, which can be explained by the higher number of demonstration data in those areas. Table 1 shows the results of our experiment on the set of AntMaze map where we denote "w/aug." in the case of stitching data augmentation and "w/o aug" otherwise. Also, we remind that for HIQL, "w/ repr." means the use of reprensentations for subgoals and goals, in opposite of "w/o repr." that takes raw values (see Park et al. (2024)). We see that our method is near the state-of-the art on smaller maps, only beaten by TT on the AntMaze-Medium maps. QPHIL outperforms all other methods on the larger maps, reaching 70% success-rate on AntMaze-ultra which is the best to our knowledge

on this benchmark, beating HIQL by at least $10\%$ on average on top of having a lower standard deviation.

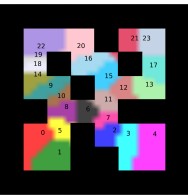
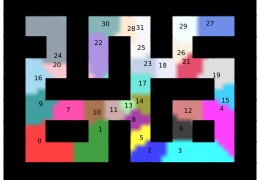
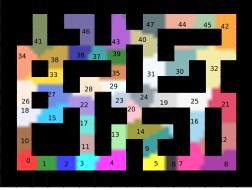

(a) Antmaze Medium Diverse          (b) Antmaze Large Diverse          (c) Antmaze Ultra Diverse

Figure 5: **Example of tokenizations using antmaze environements.** Each token has an associated color. The color of each point in the background corresponds to the color of the associated token.

## 5.3 CAN QPHIL HANDLE SPARSE DATA SCENARIOS USING TOKEN-LEVEL STITCHING ?

To test QPHIL's scaling ability in more difficult settings, we created a larger AntMaze environment called AntMaze-Extreme (figure 6a), along with two datasets variants "diverse" and "play". As shown in Figure 6, in AntMaze-Extreme QPHIL attains up to $50\%$ success rate, which is significantly above baseline results, e.g. HIQL scores $21.9\%$ and $22.8\%$ in diverse and play datasets, respectively. While QPHIL remains competitive in short-term settings, our approach is especially well suited for long-distance goal reaching navigation, in which the explicit stitching strategy afforded by our space discretization is crucial.

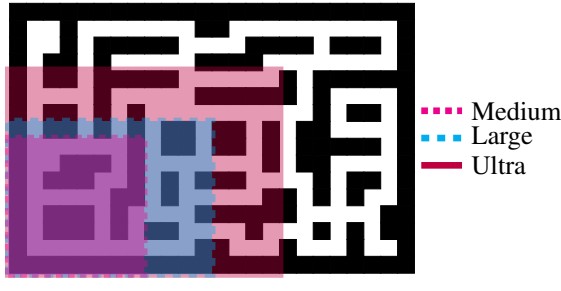

(b) Performances on Antmaze-Extreme.

| Method | Diverse | Play |
|---|---|---|
| GCBC | 13.5 ±9.5 | 12.3 ±2.1 |
| GC-IQL | 6.5 ±6.1 | 8.0 ±4.8 |
| HGCBC | 18.5 ±6.9 | 14.5 ±7.0 |
| HIQL w/repres. | 13.7 ±5.6 | 14.2 ±7.4 |
| HIQL w/o repres. | 21.9 ±6.6 | 22.8 ±7.9 |
| QPHIL w/aug. | **39.5 ±13** | **50.0 ±6.9** |
| QPHIL w/o aug. | 12.5 ±8.5 | 40.5 ±9.4 |

(a) AntMaze-Extreme topview.

Figure 6: **Evaluating QPHIL on AntMaze-Extreme.** (left) A top view of the AntMaze-Extreme map with size comparison. (right) QPHIL is outperforming all tested benchmarks in larger settings.

## 5.4 WHAT IS THE IMPACT OF THE CONTRASTIVE LOSS USED FOR LANDMARK LEARNING ?

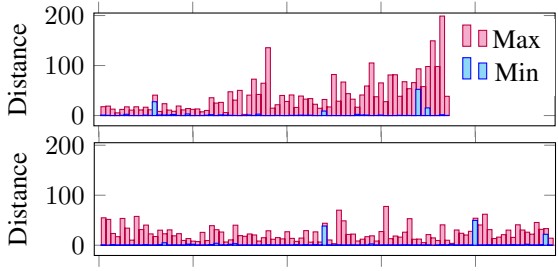

(a) Performances on Antmaze-Extreme.

| Method | Diverse | Play |
|---|---|---|
| QPHIL (non-cont) w/aug. | 16.5 ±0.9 | 26.2 ±1.5 |
| QPHIL (non-cont) w/o aug. | 8.3 ±4.7 | 16.0 ±9.3 |
| QPHIL w/aug. | **39.5 ±13** | **50.0 ±6.9** |
| QPHIL w/o aug. | 12.5 ±8.5 | 40.5 ±9.4 |

Figure 7: **Non-contrastive (top) and contrastive (bottom) intertoken distance histograms.** We see that the non-contrastive tokenization (green) results in higher extreme sized tokens while the contrastive tokenization (blue) yields a soother repartition.

The use of a contrastive loss is essential in the context of high dimensional data, where the VQ-VAE reconstruction loss is not enough to learn temporally consistent latent encodings (meaning that temporally nearby states share spatially nearby encodings).

To assess the impact of the contrastive loss on the learned landmarks, we compute, for each token, the minimum and maximum distances between states position and their corresponding codebook's decoded position. This is represented as histograms in Figure 7. While the unconstrained VQ-VAE tends to allocate higher token density in areas of higher data density, we observe that the contrastive loss results in a smoother repartition of the tokens, which in consequence increases the performance of our model. This allows to stabilize conditonning in areas of high data density. Further analysis is provided in appendix D.8.

### 5.5 DOES QPHIL STILL PERFORMS IN DIVERSE STATE-TARGET INITIALIZATIONS ?

Table 2: **Performance in long-range Random-AntMaze environments.** QPHIL is robust to random start and goal initializations, maintaining high success rates.

| Dataset | HIQL w/ repr. | HIQL w/o repr. | QPHIL w/ aug. | QPHIL w/o aug. |
|---|---|---|---|---|
| r-ultra play | 47.8 ±6.7 | 52.8 ±4.9 | 62.5 ±4.2 | **62.8** ±7.5 |
| r-ultra diverse | 45.8 ±6.2 | 49.6 ±4.7 | 58.0 ±6.6 | **62.2** ±4.8 |
| r-extreme play | 19.0 ±3.3 | 21.3 ±5.9 | **41.5** ±7.2 | 34.8 ±7.2 |
| r-extreme diverse | 23.8 ±5.8 | 23.1 ±3.7 | **45.0** ±4.5 | 31.2 ±6.9 |

Previous sections relied on evaluating performance in AntMaze environments by classically sampling initial states $s_0$ and goals $g$ from narrow distributions near two fixed points, requiring the agent to cross the entire maze. Regarding QPHIL, given such state and goal distributions do not span across multiple learned landmarks, each sampled navigation scenario leads to the similar landmark-based conditioning for our low-level policy, which is not convenient to conduct a comprehensive performance evaluation. Consequently, we designed Random-AntMaze evaluation environments, which cycles through a diverse set of 50 couples of $(s_0, g)$ allowing a more rigorous test of the generalization capabilities of our model. We refer the reader to Figure 11 in appendix for visualizations.

Table 2 showcases the performance of HIQL and QPHIL on Random-AntMaze-Ultra and Random-AntMaze-Extreme. In the random initialization setting QPHIL still performs significantly better than HIQL, reaching up to 20% improvement in success rate, which is the previous best method to our knowledge.

## 6 CONCLUSION

We proposed QPHIL, a hierarchical offline goal-conditioned reinforcement learning method that leverages pre-recorded demonstration to learn a discrete representation and temporally consistent representation of the state space. QPHIL utilizes those discrete state representations to plan subgoals in a discretized space and guide a low policy towards its final goal in a human-inspired manner. QPHIL reaches as a result top performance in challenging long-term navigation benchmarks, showing promising next steps for discretization and planning in continous offline RL settings.

**Limitations and next directions** QPHIL is a method aimed at navigation as it aims to solve tasks where a low amount of landmarks is sufficient. An interesting direction would be to analyse how to leverage such discrete representation in more intricate planning settings, such as multi-task robotics scenarios. Also, while leveraging the planning capabilities of QPHIL for model predictive control usecases could be an interesting follow up, enhancing the learning of the landmarks to adapt their size or spread regrading the uncertainty or the importance that characterize each area also constitute very promising next steps.

## 7 REPRODUCIBILITY STATEMENT

To ensure the reproductibility of our experiments, we provided all needed implementation details and hyperparameter values in the appendix. All our training have been seeded with the same 8 seeds: 0, 1, 2, 3, 4, 6 and 7. Also, we provide QPHIL's codebase along with the pretrained models ready to be evaluated on our test environments.

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

## A  DETAILS ABOUT BASELINES

**Implicit Q-Learning (IQL)**   The main issue of offline RL is the overestimation of the value of out-of-distribution actions when minimizing the temporal difference error, leading the policy to favor overestimated actions:

$$\mathcal{L}_{TD} = \mathbb{E}_{(s_t,a_t,s_{t+1})\sim\mathcal{D}} \left[ (r(s_t,a_t) + \gamma \max_{a_{t+1}} Q_{\hat{\theta}_Q}(s_{t+1},a_{t+1}) - Q_{\theta_Q}(s_t,a_t))^2 \right] \tag{3}$$

with $r(s_t,a_t)$ the task reward, $\hat{\theta}_Q$ the parameters of a target network. Without further interactions with the environment, those over-estimations cannot be corrected. While other work propose to regularize the loss function to avoid sampling out-of-distribution actions, Kostrikov et al. (2021) proposes IQL which estimates the in-distribution $\max Q$ operator with expectile regression. To do so, it learns a state value function $V_{\theta_V}(s_t)$ along a state-action value function $Q_{\theta_Q}(s_t,a_t)$:

$$\mathcal{L}_V(\theta_V) = \mathbb{E}_{(s_t,a_t)\sim\mathcal{D}} \left[ L_2^\tau(Q_{\hat{\theta}_Q}(s_t,a_t) - V_{\theta_V}(s_t)) \right] \tag{4}$$

$$\mathcal{L}_Q(\theta_Q) = \mathbb{E}_{(s_t,a_t,s_{t+1})\sim\mathcal{D}}[r(s_t,a_t) + \gamma\, V_{\theta_V}(s_{t+1}) - Q_{\theta_Q}(s_t,a_t))^2] \tag{5}$$

with $L_2^\tau(u) = |\tau - \mathbb{1}(u<0)|u^2, \tau \in [0.5,1)$ the expectile loss, which corresponds to an asymmetric square loss which penalises positive values more than negative ones the more $\tau$ tends to 1, consequently leading $V_{\theta_V}(s_t)$ to lean towards $\max_{a_{t+1}\in\mathcal{A},\, \text{st. } \pi_\beta(a_t|s_t)>0} Q_{\hat{\theta}_Q}(s_t,a_t)$ with $\pi_\beta$ the datasets behavior policy. The use of two different networks is justified to train the value function only on the dataset action distribution without incorporating environment dynamics in the TD-error loss, which avoids the overestimation induced by lucky transitions. Then, the trained $V_{\theta_V}(s_t)$ and $Q_{\theta_Q}(s_t,a_t)$ are used to compute advantages to extract a policy $\pi_{\theta_\pi}$ with advantage weighted regression (AWR):

$$\mathcal{L}_\pi(\theta_\pi) = -\mathbb{E}_{(s_t,a_t)\sim\mathcal{D}} \left[ \exp\left( \beta(Q_{\hat{\theta}_Q}(s_t,a_t) - V_{\theta_V}(s_t)) \right) \log \pi_{\theta_\pi}(a_t|s_t) \right], \tag{6}$$

with $\beta \in (0,+\infty]$ an inverse temperature. This corresponds to the cloning of the demonstrations with a bias towards actions that present a higher Q-value.

**Hierarchical Implicit Q-Learning (HIQL)**   In the offline GCRL setting, the rewards are sparse, only giving signals for states where the goal is reached. Hence, as bad actions can be corrected by good actions and good actions can be polluted by bad actions in the future of the trajectory, offline RL methods are at risk of wrongly label bad actions as good one, and conversely good actions as bad ones. This leads to the learning of a noisy goal-conditioned value function: $\hat{V}(s_t,g) = V^*(s_t,g) + N(s_t,g)$ where $V^*$ corresponds to the optimal value function and $N$ corresponds to a noise. As the 'signal-to-noise' ratio worsens for longer term goals, offline RL methods such as IQL struggle when the scale of the GCRL problem increases, leading to noisy advantages in IQL's AWR weights for which the noise overtakes the signal. To alleviate this issue, Park et al. (2024) propose HIQL which leverages a learned noisy goal conditioned action-free value function inspired by IQL:

$$\mathcal{L}_V(\theta_V) = \mathbb{E}_{(s_t,s_{t+1})\sim\mathcal{D},g\sim p(g|\tau)} \left[ L_2^\tau(r(s_t,g) + \gamma V_{\hat{\theta}_V}(s_{t+1},g) - V_{\theta_V}(s_t,g)) \right] \tag{7}$$

by using it to train two policies, $\pi^h(s_{t+k}|s_t,g)$ to generate subgoals from the goal and $\pi^l(a_t|s_t,g)$ to generate actions to reach the subgoals:

$$\mathcal{L}_{\pi^h}(\theta_h) = \mathbb{E}_{(s_t,s_{t+k},g)}[\exp(\beta\cdot(V_{\theta_V}(s_{t+k},g) - V_{\theta_V}(s_t,g)))\log\pi^h_{\theta_h}(s_{t+k}|s_t,g)] \tag{8}$$

$$\mathcal{L}_{\pi^l}(\theta_l) = \mathbb{E}_{(s_t,a_t,s_{t+1},s_{t+k})}[\exp(\beta\cdot(V_{\theta_V}(s_{t+1},s_{t+k}) - V_{\theta_V}(s_t,s_{t+k})))\log\pi^l_{\theta_l}(a_t|s_t,s_{t+k})] \tag{9}$$

With this division, each policy benefits from higher signal-to-noise ratios as the low-policy only queries the value function for nearby subgoals $V(s_{t+1},s_{t+k})$ and the high policy queries the value function for more diverse states leading to dissimilar values $V(s_{t+k},g)$. For high-dimensional states like images, HIQL proposes to learn states and goal representations $\phi(s)$ to reduce the dimension, allowing consequently easier subgoal generation. "w/ repr." and "w/o repr." refer to the variants of the HIQL with and without representations respectively. With representations, the writing of the policies become $\pi^h_{\theta_h}(\phi(s_{t+k})|s_t,g)$ and $\pi^{low}_{\theta_l}(a_t|s_t,\phi(s_{t+k}))$. We display the two in our tables for a better comparison.

**Flat-policies "signal-to-noise" issues** The usual policy learning strategy in offline RL and as such goal-conditioned offline RL is to learn the policy by weighting its updates by a function of is goal-conditioned advantage. In the case of HIQL, low-level and high-level policy updates are performed using a learned action-free advantage function: $\hat{A}(s_t, s_{t+1}, g) = \hat{V}(s_{t+1}, g) - \hat{V}(s_t, g)$, where $\hat{V}$ itself is a neural network learned through action-free IQL updates. As high values corresponds to an expectation on the discounted cumulative sum of rewards, given a state and a goal, they indicate a notion of temporal proximity in the goal-conditioned sparse rewards case. The advantage gives consequently an indication of an approach of the goal $g$ allowing the policy to learn directions. However, as the goal moves away from the current state, the learned value function may provide noisy estimates. We can write the learned value function in the from: $\hat{V}(s, g) = V^*(s, g) + N(s, g)$ where $V^*(s, g)$ corresponds to the minimal (in-distribution) distance between $s$ and $g$ and $N(s, g)$ a random noise. Borrowing from the HIQL paper, we could assume $N(s, g) = \sigma z_{s,g} V^*(s, g)$, where $\sigma$ is the standard deviation and $z_{s,g}$ is the a random variable that follows a standard normal distribution. This corresponds to a gaussian noise proportional to the temporal distance between $s$ and $g$. If we rewrite the advantage:

$$\hat{A}(s_t, s_{t+1}, g) = \hat{V}(s_{t+1}, g) - \hat{V}(s_t, g) \tag{10}$$

$$= V^*(s_{t+1}, g) + \sigma z_{s_{t+1},g} V^*(s_{t+1}, g) - V^*(s_t, g) + \sigma z_{s_t,g} V^*(s_t, g) \tag{11}$$

$$= V^*(s_{t+1}, g) - V^*(s_t, g) + \sigma z_{s_{t+1},g} V^*(s_{t+1}, g) - \sigma z_{s_t,g} V^*(s_t, g) \tag{12}$$

$$\stackrel{d}{=} \underbrace{A^*(s_{t+1}, s_t, g)}_{signal} + \underbrace{z\sqrt{\sigma_1^2 V^*(s_{t+1}, g)^2 + \sigma_2^2 V^*(s_t, g)^2}}_{noise} \tag{13}$$

Hence, for faraway goals, the "signal-to-noise" ratio (SNR) defined in this case by:

$$SNR(s_{t+1}, s_t, g) = \frac{A^*(s_t, s_{t+1}, g)}{N(s_t, s_{t+1}, g)} = \frac{A^*(s_t, s_{t+1}, g)}{z\sqrt{\sigma_1^2 V^*(s_{t+1}, g)^2 + \sigma_2^2 V^*(s_t, g)^2}}$$

can be underwhelming, because the difference in advantage is too small compared to the noise induced by the estimation. HIQL proposes to learn two different policies though the advantage estimates of a single value function. A high policy $\pi^h(s_{t+k}|s_t, g)$ is trained to generate find subgoals that maximizes $\hat{V}(s_{t+k}, g)$ and a low policy $\pi^l(a_t|s_t, g)$ is trained to maximize $\hat{V}(s_{t+1}, g)$. Hence, we can write the SNR for each level:

$$SNR^h(s_t, s_{t+k}, g) = \frac{A^*(s_t, s_{t+k}, g)}{N(s_t, s_{t+k}, g)} = \frac{A^*(s_t, s_{t+k}, g)}{z\sqrt{\sigma_1^2 V^*(s_{t+k}, g)^2 + \sigma_2^2 V^*(s_t, g)^2}}$$

$$SNR^l(s_t, s_{t+1}, s_{t+k}) = \frac{A^*(s_t, s_{t+1}, s_{t+k})}{N(s_t, s_{t+1}, s_{t+k})} = \frac{A^*(s_t, s_{t+1}, s_{t+k})}{z\sqrt{\sigma_1^2 V^*(s_{t+1}, s_{t+k})^2 + \sigma_2^2 V^*(s_t, s_{t+k})^2}}$$

The division in two policies allows to increase the high-policy SNR by comparing values from more distance states (higher signal) while also increasing the low-policy SNR by decreasing the distance between state and goal (lower noise). However, as the low-policy has to generate an instant action information and as the high-policy has to be conditioned on the real goal, HIQL seeks to find the optimal step $k$ that balanced both $SNR^h$ and $SNR^l$, which might still pose scaling issues for longer-range settings. Consequently, in very long-range scenarios, the high policy $\pi^h(s_{t+k}|s_t, g)$ may lead to noisy subgoal estimates, varying at each timestep and as such difficult for the low policy to follow.

# B  IMPLEMENTATION DETAILS

---

**Algorithm 1** Navigation with QPHIL

---

**Require:** Goal state $g$, Start state $s_0$, State quantizer $\phi$, Sequence generator $\pi^{plan}$, Navigation policies $\pi^{\text{landmark}}$ and $\pi^{\text{goal}}$

    $\tau^< = (\phi(s_0)); \tau^> = \{\}; s = s_0;$
    **while** Not exceeding a max number of steps **do**
        **if** First step (or if re-planning is required) **then**               ▷ Generate future landmarks
            $\tau^> \leftarrow (\omega_1, \omega_2, \ldots, \dot\omega) \sim \pi^{\text{plan}}(. \mid \phi(g), \tau^<)$
        **if** $\tau^>_0 \mathbin{!=} \dot\omega$ **then** $a \sim \pi^{\text{landmark}}(. \mid s, \tau^>_0)$
        **else** $a \sim \pi^{\text{goal}}(. \mid s, g)$
        Emit $a$ in the environment and observe new state $s'$
        $s \leftarrow s';$
        **if** $\tau^>_0 \mathbin{!=} \dot\omega$ **then**
            **if** $\phi(s) == \tau^>_0$ **then**               ▷ Subgoal reached, go to the next one
                $\tau^<.\text{append}(\tau^>.\text{pop}(0));$
        **else**
            **if** $s$ is in $g$ **then**                    ▷ Final goal reached
                return;

---

## B.1  IMPLEMENTATION AND TRAININGS

Our implementation of QPHIL is based on the Pytorch (Paszke et al. (2019)) machine learning library. It is available in the supplementary materials as a .zip file and will be next available in a public dedicated repository. We ran our experiments on a GPU cluster composed of Nvidia V100 GPUs, each training taking approximately 10 hours in the right conditions.

## B.2  HYPER-PARAMETERS

We present bellow the architectural choices and the hyper-parameters used to produce the results presented in Table 1 and Figure 6.

**VQ-VAE**  For the VQ-VAE, we based our implementation on the https://github.com/lucidrains/vector-quantize-pytorch.git repository. We utilize for all AntMaze variants as the encoder and the decoder a simple 2-layer MLP with ReLU activations and hidden size of 16 and latent dimension of 8. For antmaze, we add a gaussian noise to the input positions and we normalize the positions before feeding them to the encoder. Also, we vary the number of encodings in the codebook as the map grows. We provide bellow the complete list of used hyper-parameters in Table 3. Unless specified, the VQ-VAE hyper-parameters are the defaults ones from the initial library's implementation.

**Transformer**  For the transformer, we used as described an encoder-decoder architecture inspired by the paper Vaswani et al. (2017) provided by the **torch.nn library**. We use a max sequence length of size 128, an embedding dimension of 128, a feed-forward dimension of 128, 4 layers of 4 heads and a dropout of 0.2. We train our model with 250 epochs when we apply stitching and 2500 epochs otherwise. We perform validation computation with a 0.95 dataset split and perform sampling with a temperature of 0.9. We optimize our model using the Adam optimizer with a learning rate of 1e-5. All of those results are also presented in Table 4 bellow.

**Low subgoal policy**  For the low subgoal policy and its value function, we adapt the policy proposed by HIQL. For the value function, we use a MLP policy of 3 layers with hidden size of 512 and GeLU activations. We apply layer normalization for each layers and initialize the weights with a variance scaling initialization of scale 1. No dropout is used for the value function. For the policy, we use a two layer MLP with hidden size of 256 and ReLU activations. We don't apply layer norm and initialize the parameters though a variance scaling initialization of scale 0.01. Our policy

Table 3: VQ-VAE hyper-parameters

| Hyper-parameter | Value |
| --- | --- |
| Epochs | 1000 |
| Batch size | 16384 |
| Contrastive coef | 2e1 |
| Commit coef | 1e3 |
| Reconstruction coef | 1e5 |
| Norm coef | [22.5,22.5] (medium), [40,30] (large) [55,40] (ultra), [90,50] (extreme) |
| Noise standard deviation | 2.0 |
| Quantizer window | 100 |
| Latent dim | 8 |
| Encoder and decoder hidden dims | 16 |
| Codebook size | 24 (medium), 32 (large), 48 (ultra), 96 (extreme) |
| Learning rate | 3e-4 |

Table 4: Transformer hyper-parameters

| Hyper-parameter | Value |
| --- | --- |
| Epochs | 250 (stitching), 2500 (no stitching) |
| Batch size | 64 |
| Validation split | 0.95 |
| Max sequence length | 128 |
| Embedding dim | 128 |
| Num layers | 4 |
| Num heads | 4 |
| Dropout | 0.2 |
| Sampling temperature | 0.9 |
| Learning rate | 1e-5 |

outputs the mean and standard deviation of an independent normal distribution. We clamp the log std of the output between -5 and 2. We train both the value function and the policy at the same time, performing 1e6 gradient steps with a batch size of 1024. For the IQL parameters, $\beta = 3.0$, the expectile $\tau = 0.9$, the polyak coefficient is 0.005 and the discount factor $\gamma = 0.995$. We clip the AWR weights to 100. Also, we sample the next token with a probability of 0.8 and the current token with probability 0.2. The targets updates are performed at each gradient step. We summarize bellow in Table 5 the given hyper-parameters:

Table 5: Low policy and low value hyper-parameters

| Hyper-parameter | Value |
| --- | --- |
| Gradient steps | 1e6 |
| MLP num layers | 3 (value function), 2 (policy) |
| MLP hidden sizes | 512 (value function), 256 (policy) |
| Activations | GeLU (value function), ReLU (policy) |
| Layer normalization | True (value function), False (policy) |
| Variance scaling init scale | 1 (value function), 0.01 (policy) |
| Policy log std min and max | -5 and 2 |
| Batch size | 1024 |
| $\beta$ | 3.0 |
| Expectile $\tau$ | 0.9 |
| Polyak coef | 0.005 |
| Discount factor $\gamma$ | 0.995 |
| Clip score | 100 |
| $p_{future}$ | 0.8 |
| $p_{current}$ | 0.2 |

**Low goal policy**  For the low goal policy, we trained our GC-IQL implementation in Pytorch with GC-IQL's hyper-parameters taken the HIQL paper: $(\beta, \tau, \gamma) = (0.99, 3, 0.9)$ for the smaller mazes (AntMaze-{Medium-Large}) and $(\beta, \tau, \gamma) = (0.995, 1, 0.7)$ for the larger mazes (AntMaze-{Ultra,Extreme}).

**Baselines**  For all the AntMaze environment except for the Extreme and Random variants, the reported results are the ones provided in the HIQL paper. For the extreme variant, we performed a grid search on the official HIQL implementation with $\beta \in \{1, 3, 10\}$, $\tau \in \{0.7, 0.9\}$, $\gamma \in \{0.99, 0.995\}$ and $k \in \{25, 50, 75, 100\}$ across 4 seeds (0, 1, 2, 3). We found that for extreme the best hyper-parameter set was: $(\beta, \tau, \gamma, k) = (10, 0.7, 0.995, 100)$ for HIQL. We used this set for the training of HIQL on the AntMaze-Extreme variants and used $(\beta, \tau, \gamma) = (0.995, 1, 0.7)$ for GC-IQL. For GCBC, we launched or trainings with the same hyper-parameters as ultra from the HIQL paper. For HGCBC, we performed an hyper-parameter search on $k \in \{25, 50, 75, 100\}$ across 8 seeds and found no significantly better set, so we took 100 which is the same as HIQL.

B.3   DATA CLEANING

Instability in some trajectories of the dataset or an indecisive tokenizer can lead to wobbliness in sequences of the dataset. For instance, successive states that are close to two tokens could lead to alternating tokens in the sequence. While this gives information, it would make training harder for the sequence generator. To correct this, if a state has already been seen during the last for steps, it is not taken into account. Cycles can appear in sequences, especially with data augmentation. Cycles in the dataset create two problems: the model would learn to generate cycles which leads to indefinite repetitions of cycles at inference; and cycles create longer episodes that tends to reduce the overall performance of the model. To correct this, we remove cycles from the dataset. Finally, the finite number of tokens leads to repeated sequences, a problem that is exacerbated by data augmentation. To prevent the model from learning to repeat the same sequences, we remove repeated sequences from the dataset. This is efficiently implemented using a hash table.

---

**Algorithm 2** Sequence Cleanup: Removing Wobbly Tokens, Cycles, and Repeated Sequences

---

**Require:** Dataset of tokenized sequences $\mathcal{D} = \{S_i\}_{i=1}^{N}$, where each $S_i = \{z_1^i, z_2^i, \ldots, z_{T_i}^i\}$
**Ensure:** Cleaned dataset $\mathcal{D}_{\text{clean}}$
   Initialize cleaned dataset $\mathcal{D}_{\text{clean}} \leftarrow \emptyset$
   Initialize a hash table $H$ to store unique sequences
   **for** each sequence $S_i$ in $\mathcal{D}$ **do**
      Initialize $S_{\text{temp}} \leftarrow \emptyset$                    ▷ Temporary sequence for filtering tokens
      Initialize history history $\leftarrow \emptyset$        ▷ Track recent tokens to prevent wobbliness
      **for** each token $z_t^i$ in sequence $S_i$ **do**
         **if** $z_t^i$ has not been seen in the last 4 steps **then**
            Append $z_t^i$ to $S_{\text{temp}}$
            Update history to include $z_t^i$
      **Remove Cycles:** Detect and remove any cycles in $S_{\text{temp}}$
      Initialize an empty set visited
      **for** each token $z_t^i$ in $S_{\text{temp}}$ **do**
         **if** $z_t^i$ is in visited **then**
            Truncate $S_{\text{temp}}$ up to the first occurrence of $z_t^i$ to remove the cycle
            **break**
         **else**
            Add $z_t^i$ to visited
      **Remove Repeated Sequences:**
      **if** $S_{\text{temp}}$ is not in the hash table $H$ **then**
         Add $S_{\text{temp}}$ to the cleaned dataset $\mathcal{D}_{\text{clean}}$
         Insert $S_{\text{temp}}$ into the hash table $H$
     **return** Cleaned dataset $\mathcal{D}_{\text{clean}}$

---

### B.4 TOKENIZATION PIPELINE

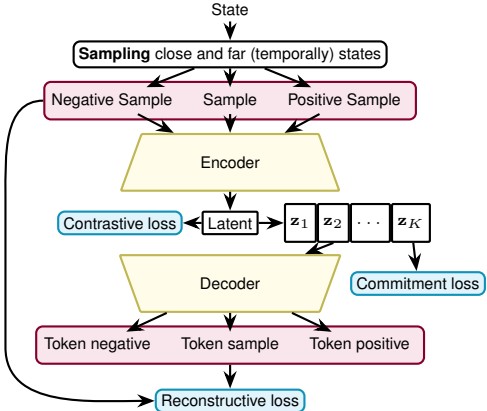

Figure 8: Pipeline used to learn the tokenization.

### B.5 TOKEN SEQUENCE COMPRESSION

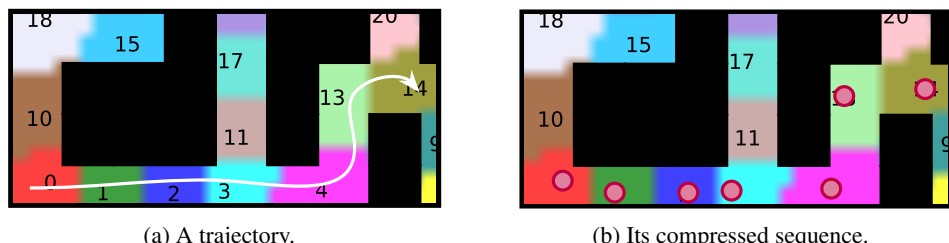

(a) A trajectory.  (b) Its compressed sequence.

Figure 9: Sequence compression process. Sequences of tokens are simplified by removing repetitions of the same token, giving a minimal representation of sequences in terms of token.

### B.6 TRAJECTORY STITCHING

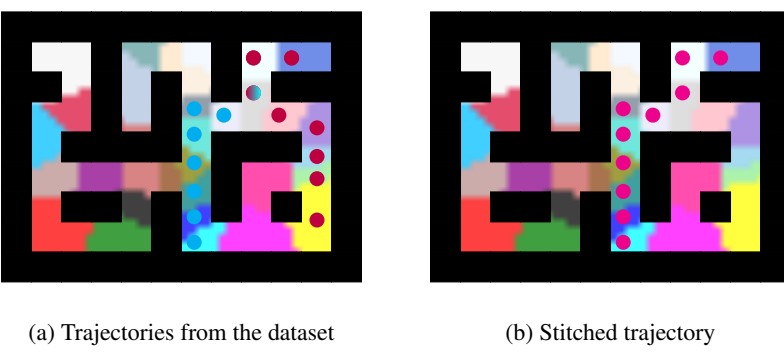

(a) Trajectories from the dataset  (b) Stitched trajectory

Figure 10: Illustration of the data augmentation process.

# C  DETAILS ABOUT ENVIRONMENTS AND DATASET

## C.1  RANDOM-ANTMAZE VISUALISATIONS

To further test the capabilities of our planner, we tested our approach on a new version of the Antmaze environments that include sampling circularly from a finite set of 50 random $(s_0, g)$ positions. We show bellow the $(s_0, g)$ distributions along with associated sampled planned trajectories of our transformer for the Antmaze and Random-Antmaze variants of the environments.

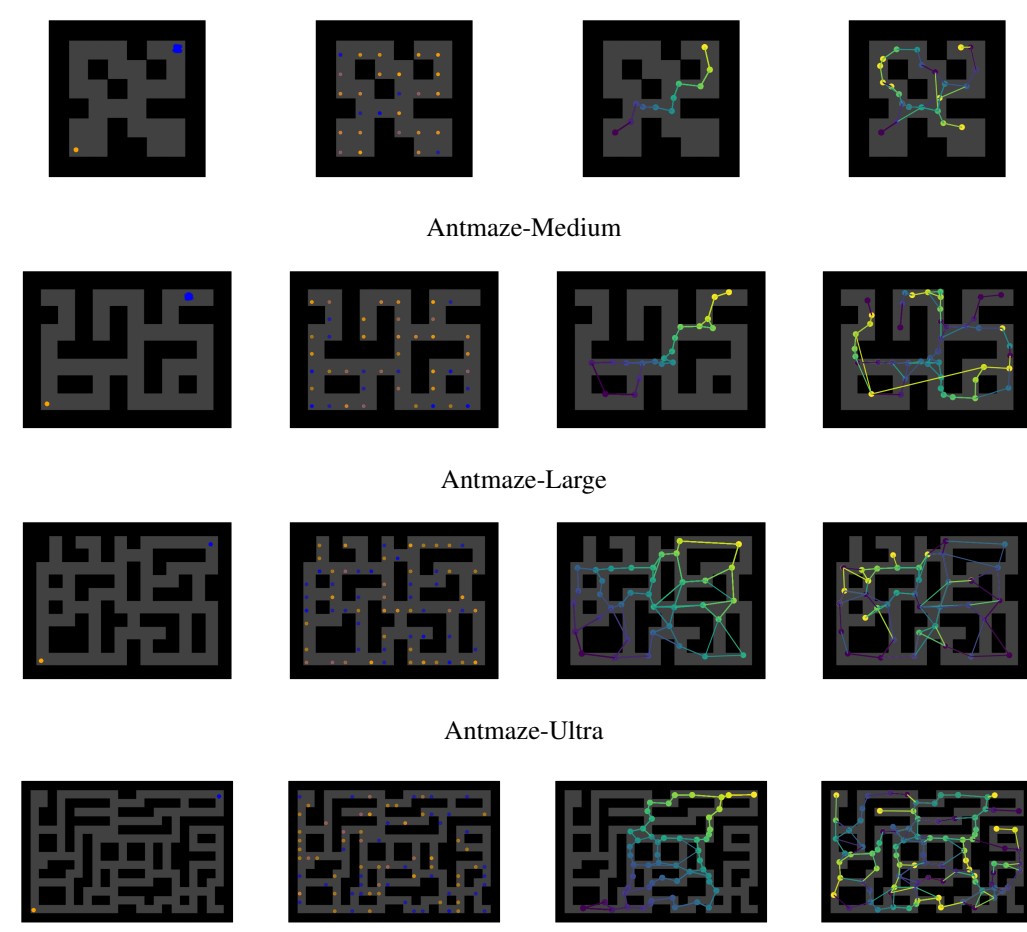

Antmaze-Medium

Antmaze-Large

Antmaze-Ultra

Antmaze-Extreme

Figure 11: **Initializations comparisions between AntMaze and Random-AntMaze.** (left) Plots of 50 sampled starting positions $s_0$ (in blue) and target goals $g$ (in orange) for AntMaze and Random-Antmaze. We see that Random-AntMaze has a broader $s_0, g$ distribution and as such is a better fit for a comprehensive evaluation of navigation tasks. (left) Plots of 50 sampled planing paths with a color gradient indicating the order in sequence (yellow to blue). The high policy subgoals exhibit higher diversity in the Random-AntMaze variations.

### C.2 ANTMAZE-EXTREME DATASETS

AntMaze is a very popular benchmark in offline reinforcement learning, and it is part of the D4RL Fu et al. (2020) dataset suite, interfaced through the Gym library Brockman et al. (2016). It uses the Mujoco physics engine Todorov et al. (2012) for its simulation.

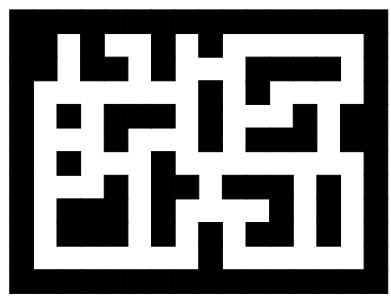

Figure 12: Maze of Antmaze ultra

The antmaze environment consist of a maze-like structure in which a simulated ant agent must navigate from a starting position to a goal position. Contrary to what one might expect, the agent is not controlled by simple directional commands. Instead, the ant is controlled through an 8-dimensional continuous action space. Due to the complex dynamics of the ant and the intricate structure of the maze, this environment poses a significant challenge for both exploration and planning. Each dimension of the action space corresponds to a torque applied to one of the ant's joints. Values in the action space are bounded between -1 and 1 and are in Nm. Hence, $\mathcal{A} = [-1, 1]^9$. The observation space is a 29-dimensional continuous space corresponding to the cartesian product of the x,y ant coordinates and the ant's configuration space $S_{ant}$. This space is unbounded in all directions: $\mathcal{S}_{ant} = \mathbb{R}^{27}$. The first dimension is the height in meter of the torso. The four following dimensions correspond to respectively the $x$, $y$, $z$ and $w$ orientation in radian of the torso. The height next dimensions are the angles between different links, in radian. Then, $x$, $y$ and $z$ velocities in m.s$^{-1}$ followed by their respective angular velocities and angular velocities of all links, in rad.s$^{-1}$. The goal space is a subspace of $\mathbb{R}^2$: goals are given in $x$, $y$ coordinates. The reward is sparse: it is equal to 0 until the ant has reach the goal, where it is equal to 1. Hence, $\mathcal{R} = \{0, 1\}$.

There are three variations of this environment: medium, large and ultra; each one consisting of a different maze structure. The ultra variant is not part of the original dataset and has been introduced in Jiang et al. (2022). Each maze has a different datasets, each one composed of 1000 trajectories of 1000 steps, for a total of $10^6$ steps. D4RL provides two variations datasets per: "play" and "diverse". The former is generated using hand-picked locations for the goal and the starting position whereas the latter is generated using a random goal position and starting position for each trajectory.

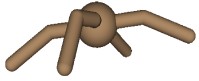

Figure 13: Ant

The extreme maze was designed following the same principles as the original maps. Its surface area is approximately 166% larger than antmaze ultra and three times the size of antmaze large. This new map is a direct extension of the original implementation, and both the implementation and the datasets are provided. A comparison is available in Figure 7a. The two provided datasets are "play" and "diverse", both collected using the same methods as for the smaller maps. The maze is structured as a grid, where trajectories are generated on the grid, and a trained policy follows these paths to gather data.

# D ADDITIONAL EXPERIMENTS

This sections contains additional experiments assessing the following questions:

1. Does reinforcement learning matter for the high-policy in antmaze ?
2. What is the impact of going from continuous to discrete planning ?
3. What about replanning and a closed loop format ?
4. What is the impact of the different losses when training the quantizer ?
5. How many more trajectories are generated by data augmentation ?
6. How does the coverage of the dataset (in the state space) broadly impact both the learning of the VQ-VAE and the policy ?
7. Could we use a uniform discretization of the antmaze maps ?
8. What is the impact of the contrastive loss used for landmark learning ?

## D.1 DOES RL MATTER FOR THE HIGH-POLICY IN ANTMAZE ?

As we use imitation learning in the our high-level policy, we tested the performance of HIQL with a behavior cloning high level policy to look the potential impact of losing the RL weighting. As such, we tested HIQL with a classical Behavioral Cloning learning of the high level policy for the AntMaze-Ultra variants. We obtained the results provided in Table 6. We see no significant difference in performance between the two approaches, meaning that RL in high levels does not necessarily improve performance on AntMaze settings.

Table 6: **Comparison of BC+IQL and HIQL.** We see that both methods perform without significative difference.

| Dataset | BC+IQL w/ repr. | BC+IQL w/o repr. | HIQL w/ repr. | HIQL w/o repr. |
|---|---|---|---|---|
| r-ultra play | 47.4 ±15 | 43.2 ±19 | 39.2 ±15 | 56.0 ±12 |
| r-ultra diverse | 50.8 ±11 | 51.4 ±17 | 52.9 ±17 | 52.6 ±8.7 |

## D.2 WHAT IS THE IMPACT OF GOING FROM CONTINUOUS TO DISCRETE PLANNING ?

**Planning continuous subgoals** As stated in the introduction (section 1), offline reinforcement learning methods struggle for long-distance tasks even with added hierarchy levels as the signal-to-noise ratio still degrades during subgoal generation, which can result in a noisy high-level policy and, consequently, reduced performance. To test the performance of the planning in a continuous space, as simple approach is to consider an ablated HIQL which uses behavioral cloning rather than offline reinforcement learning to learn a high-level policy (used to plan high-level goals), i.e. BC+IQL from appendix section D.1. As such, we can reuse Table 6 results and produce the comparative Table 7:

Table 7: **Comparison of BC+IQL and QPHIL.** Discrete planning outperform continous planning and the Antmaze-Ultra datasets.

| Dataset | BC+IQL w/ repr. | BC+IQL w/o repr. | QPHIL w/ aug. | QPHIL w/o aug. |
|---|---|---|---|---|
| ultra play | 47.4 ±15 | 43.2 ±19 | **64.5 ±6.8** | 61.5 ±6.2 |
| ultra diverse | 50.8 ±11 | 51.4 ±17 | 61.8 ±3.7 | **70.3 ±6.9** |

QPHIL displays better performance than BC+IQL and HIQL, assessing the benefits of using discrete planning for longer range navigation scenarios.

**Codebook size impact** As another way to assess the importance of using a finite (and small) set of discrete landmarks, We conducted an experiment to assess the impact of the codebook size on the overall performance on antmaze-ultra-diverse-v0, shown in Table 10:

Table 8: **Comparison of BC+IQL and QPHIL.** Discrete planning outperform continous planning and the Antmaze-Ultra datasets.

| Codebook size | Number of used tokens | Performance of policy |
|---|---|---|
| 1 | 1 (100 % used) | 0 |
| 8 | 8 (100 % used) | 6.0 |
| 24 | 24 (100 % used) | 52.0 |
| 48 | 48 (100 % used) | 86.1 |
| 96 | 96 (100 % used) | 72.0 |
| 256 | 57 (22 % used) | 88.0 |
| 1024 | 95 (9 % used) | 88.0 |
| 2048 | 146 (7 % used) | 82.0 |

The first column corresponds to the number of available codebook vectors of the VQ-VAE quantizer. The second column corresponds to the number of used codebook vectors. We consider that a codebook vector is used if there exists a state that projects its encoding on it and that the set of states whose encodings are projected on the same codebook vector corresponds to a landmark. We observe that after a given threshold on the number of available codebook vectors, their use percentage decreases, i.e. the quantizer do not benefit from additional codebooks to achieve a good quantization. The performance of the method appears to saturate after the threshold of 48 tokens. Regarding the antmaze ultra map, this codebook size corresponds to a good trade-off ensuring an accurate "signal-to-noise" ratio while stabilizing high-level commands.

### D.3 WHAT ABOUT REPLANNING AND A CLOSED LOOP FORMAT ?

QPHIL's high policy allows full planning in an open-loop manner and shows great performance by doing so. In the experiments reported in the main body of this paper, we considered this open-loop version (as presented in figure 3), that plans the sequence of subgoals at the start of the episode and doesn't perform any further replanning then. The tokens from the initial plan are consumed each after the other once they have been reached. However, if the low policy makes a mistake and goes into an unexpected landmark, the initial can become obsolete, more optimal paths could be consider. Moreover, it might let the agent into an out-of-distribution situation for the low level policy, targeting a landmark never seen for that situation during training. Then, one might wonder if replanning a new path from this new token in a closed-loop manner would help the policy to perform better. As such, we tested several replanning strategies and analyzed their impact on the success rate of the agent. We tested on the AntMaze-Extreme variants the impact of replanning when the obtained token is different from the next planned subgoal. To replan, we sample a given number of plans with our sequence generator $\pi^{\text{plan}}$, and finally select the shortest one among those successfully reaching the goal area. We see in table 9 that re-planning at out-of-path situations doesn't significantly impact the overall performance of our models in the AntMaze-Extreme variants. Though, some improvement is still observed with the best from 10-samples version (especially for extreme-play). More advanced re-replanning strategies are left for future work (e.g., using informed Beam-Search or MCTS decoding strategies), but this result is promising for the ability of QPHIL to deal with more complex environments (with some distribution shifts of the dynamics or with stochasticity for instance, where re-planning could look as crucial).

Table 9: **QPHIL with re-planning.**

| Dataset | QPHIL 1-sample | QPHIL 10-samples |
|---|---|---|
| extreme play | 35.5 ±7.8 | 44.3 ±16 |
| extreme diverse | 38.5 ±8.9 | 50.3 ±9.4 |

### D.4  WHAT IS THE IMPACT OF THE DIFFERENT LOSSES WHEN TRAINING THE QUANTIZER ?

In QPHIL, the VQ-VAE quantizer is learned using a linear composition of three losses: the commit loss, the contrastive loss, and the reconstruction loss, which all have an associated coefficient hyperparameter to be tuned. In our experiments, we used the same set of coefficients for each maze shape, which we found through a preliminary hyperparameter search to be robust across the diversity of maze sizes we considered. In the following, we analyze the impact of the different coefficients on the quality of the discretization.

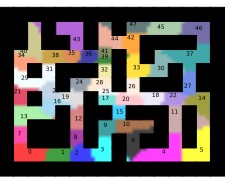 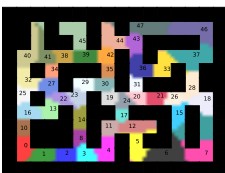 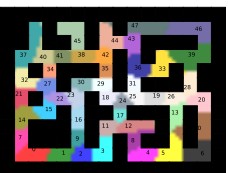 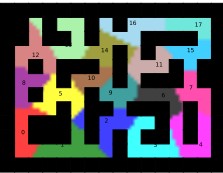

Figure 14: **Commit coefficient impact.** From left to right $\alpha_{\text{commit}} \in \{0, 1e1, 1e3, 1e6\}$, $\alpha_{\text{contrastive}} = 2e1$, $\alpha_{\text{recon}} = 1e5$. We see that a low commit coefficient leads to varying sized landmarks, while high commit loss diminishes the number of landmarks.

**Commit loss**  The commit loss serves the purpose of maintaining a vicinity between the continuous encodings and the elements of the codebook in the latent space. A zero or low commit loss creates a poor repartition of the continuous representations with regard to the codebook. This creates varying sized areas, where some are too small and some too big (e.g. figure 14, leftmost map). On the other hand, a high commit loss will force the encoder to match the codebook vectors too rigidly. This leads to information loss as the encoder might struggle to represent subtle variations in the input. Also, this increases the amount of dead codebook vectors which consequently reduces the amount of used tokens for discretization (e.g. figure 14, rightmost map).

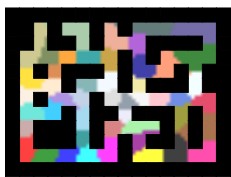 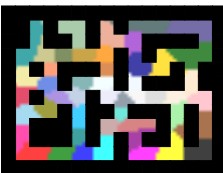 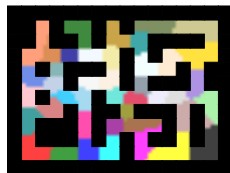 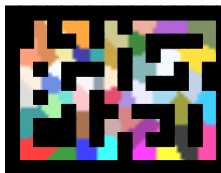

Figure 15: **Contrastive coefficient impact with reconstruction loss** With $\alpha_{\text{commit}} = 1e3$, $\alpha_{\text{contrastive}} \in \{0, 2, 2e1, 2e5\}$, $\alpha_{\text{recon}} = 1e5$. We see that a low contrastive coefficient leads to multiple piece landmarks, while high values of contrastive seem to fix this issue.

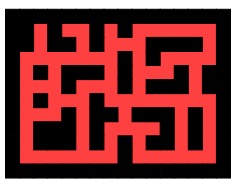 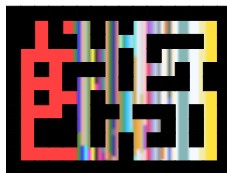 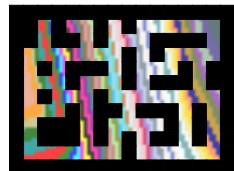 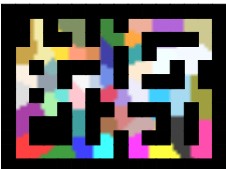

Figure 16: **Contrastive coefficient impact without reconstruction loss** With $\alpha_{\text{commit}} = 1e3$, $\alpha_{\text{contrastive}} \in \{0, 2, 2e1, 2e5\}$, $\alpha_{\text{recon}} = 0$. We see that the reconstruction loss is not needed to generate good tokens, however, it makes the tokenization more stable to hyperparamters.

**Contrastive loss**   The contrastive loss serves the purpose of organizing temporally the latent space, which is paramount to create navigable landmarks. When used jointly with the reconstruction loss (see figure 15), it ensures that the landmarks do not span through obstacles, consequently ensuring that landmarks are of single piece and navigable. If the contrastive coefficient is too low, multiple piece landmarks can appear by spanning across obstacles. If it is too high, it might overshadow other components, leading to the aforementioned failures. Used without the reconstruction loss (see figure 16), the contrastive can manage a good tokenization but the reconstruction loss helps by showing better tokenization for a higher number of contrastive coefficient values.

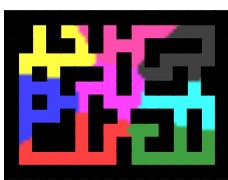 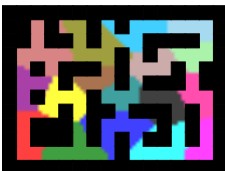 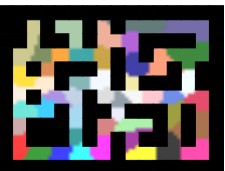 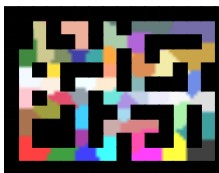

Figure 17: **Reconstruction coefficient impact without contrastive loss** With $\alpha_{\text{commit}} = 1e3, \alpha_{\text{contrastive}} = 0, \alpha_{\text{recon}} \in \{1e1, 1e3, 1e5, 1e9\}$. We see that the reconstruction loss can form a sufficient number of landmarks as long as it is not overshadowed by the commit loss. However, it fails to form fully navigable tokesn without the contrastive loss.

**Reconstruction loss**   The reconstruction loss serves the purpose of helping the contrastive loss in the learning of the tokens. Using the reconstruction loss is not necessary as shown experimentally (see figure 16, rightmost maze). However, it can serve the purpose to make the learning of the tokens more robust to variations of the other coefficients. If too low and with a small contrastive loss, the commit loss is too big and we observed a low amount of token which is coherent with the commit loss experiments (figure 14).

### D.5 HOW MANY MORE TRAJECTORIES ARE GENERATED BY DATA AUGMENTATION?

We share here the number of new **token** trajectories obtained by our stitching data augmentation on the several dataset and compare them to the initial trajectory numbers.

Table 10: **Number of token trajectories in the initial and the augmented datasets.**

| Dataset | Initial number of token trajectories | Augmented number of token trajectories |
|---|---|---|
| medium-diverse | 999.0 ±0 | 3696.5 ±298.1 |
| medium-play | 999.0 ±0 | 4092.4 ±354.4 |
| large-diverse | 999.0 ±0 | 12106.4 ±1421.4 |
| large-play | 999.0 ±0 | 12786.6 ±2578.0 |
| ultra-diverse | 999.0 ±0 | 11730.9 ±895.2 |
| ultra-play | 999.0 ±0 | 11687.8 ±1280.9 |
| extreme-diverse | 499.0 ±0 | 10203.1 ±629.0 |
| extreme-play | 499.0 ±0 | 9400.3 ±269.0 |

### D.6 HOW DOES THE COVERAGE OF THE DATASET (IN THE STATE SPACE) BROADLY IMPACT BOTH THE LEARNING OF THE VQ-VAE AND THE POLICY ?

We see experimentally that areas with very low to null coverage (specifically walls here) share the same token as one of the nearest in-distribution state, showing a certain amount of generalization of the tokenization. Also, high coverage areas have a tendency to constrain a higher number of smaller landmarks than the low coverage parts of the maze, due to the loss having more weight in those, since there are more samples. Section 5.4 illustrates that this issue is mitigated by the contrastive loss, leading to tokens of more uniform size. For the policy, as we use an offline reinforcement learning (IQL), low coverage areas are to be avoided and IQL seeks to sample actions within training distribution to avoid getting out-of-distribution.

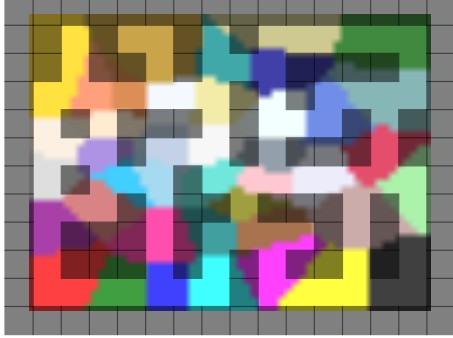

Figure 18: **Landmarks formed within the walls.**

## D.7    COULD WE USE A UNIFORM DISCRETIZATION OF THE ANTMAZE MAPS ?

In the tested Antmaze environment, it is indeed possible to discretize uniformly the maze to perform planning. However, this approach would be limited because of several points. First, it requires the user to have previous knowledge about the shape of the maze, to choose small enough lattices to avoid landmarks to span across obstacles, which is not necessarily the case and is a strong hypothesis that our method doesn't require. Also, the size of those lattices could become really small for some environments, leading to an unnecessary increase of the number of tokens, resulting in more difficult planning. Additionally, we ran experiments with uniform tokenization on the antmaze-ultra-diverse-v0 dataset with a comparable amount of tokens. This resulted in a score of $53.5 \pm 12.7$, which is 10 to 20 percent lower than with the quantizer method.

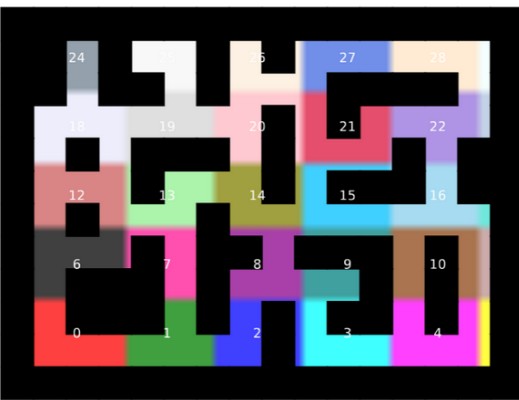

Figure 19: **Uniform tokenization on antmaze-ultra-diverse-v0.**

## D.8 WHAT IS THE IMPACT OF THE CONTRASTIVE LOSS USED FOR LANDMARK LEARNING ?

The use of a contrastive loss is paramount in the context of high dimensional data, where the VQ-VAE reconstruction loss is not sufficient to learn temporally consistent latent encodings (meaning that temporally nearby states share spatially nearby encodings). In our specific case, as we decode the positions, we observe experimentally an amount of consistency in our latent representations, as shown in Figure 20.

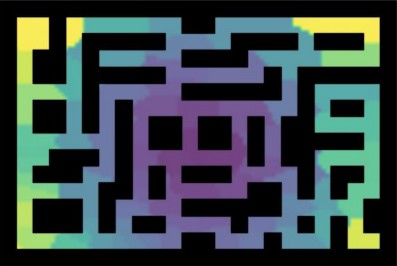

(a) Non-contrastive tokenization        (b) Contrastive tokenization

Figure 20: **Contrastive and non-constrastive latents distances from the goal.** We compute for each position the euclidiean distance between its representation and the representation of the central position. We see that the overall latents are spatially/temporally well organized, even if the non-contrastive tokenization seem to get slightly more extreme distances on the edges.

However, the learning of the unconstrained VQ-VAE tends to increase token density in high density data areas, allowing for a better average reconstruction loss. We represent in Figure 21 a comparison of the obtained tokens on Antmaze-Extreme.

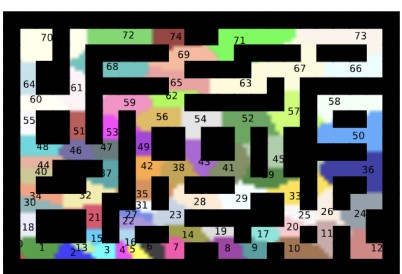

(a) Non-contrastive tokenization        (b) Contrastive tokenization

Figure 21: **Visual tokenization comparison.** The contrastive loss allows the learning of more homogenuous landmarks.

We also compute for each token the minimum and maximum distances between states position and their corresponding codebook's decoded position, represented as histograms in Figure 7. We see that the contrastive loss results in a smoother spread of the tokens, which in consequence improves the performance of our model even in the case of position decoding.

