# OpenReview forum: "Navigation with QPHIL: Offline Goal-Conditioned RL in a Learned Discretized Space"
_ICLR.cc/2025/Conference — Submitted to ICLR 2025_

### Official Review · Reviewer_mB11 · 2024-11-04

**Soundness:** 3
**Presentation:** 2
**Contribution:** 2
**Rating:** 5
**Confidence:** 4

**Summary:**

A key challenge in Offline RL is the signal-to-noise ratio, which can lead to ineffective policy updates due to inaccurate value estimates. The authors propose a hierarchical transformer-based approach that utilizes a learned quantizer to simplify the navigation process and breaks the environment states into tokens. This method decouples high-level path planning from low-level path execution, allowing for a more straightforward training of low-level policies conditioned on discrete zones (tokens). This zone-level reasoning enhances planning by facilitating explicit trajectory stitching, reducing reliance on noisy value function estimates. The authors claim that this method achieves SOTA performance in complex long-horizon navigation tasks.

**Strengths:**

S1. Provided access to a larger antmaze dataset (Antmaze-extreme) with higher complexity.

S2. Develops on HIQL (Park et al, 2024) and adds a tokenization methods and a planner policy which breaks the problem into k distinct navigation problems.

S3. Outperforms all other benchmarks in larger datasets like antmaze-extreme depicting the strength of tokenization of the state space.

S4. The code provided has good documentation.

**Weaknesses:**

W1. The paper lacks a methodology diagram describing the entire training and inference procedure, which makes the paper difficult to follow.

W2. The experiments lack other navigation environments and results are only shown on Antmaze.

W3. For the low level policy, $\pi_{\text{landmark}}$ and $\pi_{\text{goal}}$ are distinct policies which are both generating low level actions to reach to $\omega$ and $g$ respectively. The need for having two different policies is unclear and sparsely mentioned in the text.

W4. In the tokenizer, the contrastive loss penalizes temporal closeness of the states in the latent space.
W4.1. The tokenization example in Figure 4, has tokens broken into very non temporally close spaces as token 47 and 46 reach areas near token 43, and token 12 reaches near 16 and 17.
W4.2. It’s also unclear how this loss “aligns with the walls thanks to the contrastive loss.” as mentioned in the caption of Figure 4.

W5. The reason for using a transformer for sequence generation is unclear and thorough reasoning for using it would be appreciated as compared to HIQL.

W6. Does not perform well on smaller antmaze examples compared to other methods like HIQL.


Minor Nitpicks

N1. Spelling mistakes, line 444-445: “Exemple”

N2. Equations that are referred are not in the nearby pages of the reference, like line 347-348, referring to eq 7,

N3. $g$ is used both as a state encoder function, and the goal.

**Questions:**

Q1. In Figure 4, please explain the tokenization in more detail wrt. the W4. Also could you please elaborate on both the parts of W4.

Q2. Could you elaborate on W3 and explain the reason behind having two distinct low level policies. Could you show some ablations which provide empirical reasons for having $\pi_\text{goal}$.

Q3. Elaborate on the reasons for using a transformer architecture in $\pi_\text{plan}$ and ablations for the same would be helpful.

Q4. Why does QPHIL not perform above SOTA in smaller Antmaze datasets? Could changing the way of tokenization help with this?

Q5. Which is the proposed method in the paper, “QPHIL w/ aug.” or “QPHIL w/o aug.”? Having both where-in “w/ aug.” performs better in some scenarios and “w/o aug.” otherwise is confusing and elaborating on that would be helpful.

Q6. Can the prescribed method QPHIL work on other navigation tasks like Autonomous driving, and how would tokenization work for such a task?

---

> ### Author Response · Authors · 2024-12-02
> **Answer to Reviewer R-mB11 (1/2)**
>
> We sincerely thank **R-mB11** for their constructive feedback and for recognizing several key strengths of our work. We are pleased that you acknowledged our AntMaze-Extreme dataset, which underscores the robustness of our evaluation framework in longer-range navigation settings. We appreciate your recognition of how our work builds upon HIQL by incorporating tokenization methods and a planner policy that effectively breaks down the problem into k distinct navigation tasks, leading to SOTA results in long-range navigation. Lastly, we are thankful for your positive remarks regarding the comprehensive documentation of our code.
>
> # 1. For the description of the training and inference procedure
>
> > W1. The paper lacks a methodology diagram describing the entire training and inference procedure, which makes the paper difficult to follow.
>
> For the inference diagram, we kindly refer **R-mB11** to **Figure 3** which describes the inference pipeline of our model. Additionally, we understand **R-mB11** concerns about the lack of diagrams concerning the training procedure. As all the components of QPHIL are trained sequentially, we updated **subsection 4.1** with a statement to better inform of such design.
>
> # 2. On the Selection of Evaluation Benchmarks
>
> > W2. The experiments lack other navigation environments and results are only shown on Antmaze.
>
> We appreciate Reviewer **R-mB11** for highlighting this concern. Since multiple reviewers have raised this issue, we kindly direct you to the **"Answer to All Reviewers"** section, specifically the **"On the selection of evaluation benchmarks"** subsection, where we address this topic in detail.
>
> # 3. About the need of two distinct low policies
>
> > W3. For the low level policy, and are distinct policies which are both generating low level actions to reach to and respectively. The need for having two different policies is unclear and sparsely mentioned in the text.
>
> > Q2. Could you elaborate on W3 and explain the reason behind having two distinct low level policies. Could you show some ablations which provide empirical reasons for having.
>
> Our method can be understood as decoupling the long-range goal reaching task into a long-range landmark reaching task and a short-range goal reaching task. Discretizing the state space allows for easier navigation across large environments but is only meant to bring the agent to the landmark of the goal. Hence, the discrete-goal conditioned policies (planner and low level) are high performing for long range planning and navigation but are not meant to perform precise goal reaching. The continuous-goal conditioned policies, on the other hand, struggle to perform long term planning but are precise enough to reach specific near goals. Hence, by first solving the long range navigation with the discrete-goal conditioned policies and then finishing by reaching precisely the goal, we can increase overhaul navigation performance. Additionally, using a single low-level policy requires consideration of a double-conditioning featuring the absolute goal position and the next landmark to reach. We believe this approach to be bound to underperform as absolute goals will bring noise to landmark reaching.
>
> # 4. About Figure 4
> > W4. In the tokenizer, the contrastive loss penalizes temporal closeness of the states in the latent space. W4.1. The tokenization example in Figure 4, has tokens broken into very non temporally close spaces as token 47 and 46 reach areas near token 43, and token 12 reaches near 16 and 17. W4.2. It’s also unclear how this loss “aligns with the walls thanks to the contrastive loss.” as mentioned in the caption of Figure 4.
>
> > Q1. In Figure 4, please explain the tokenization in more detail wrt. the W4. Also could you please elaborate on both the parts of W4.
>
> We thank the reviewer for pointing this out. We updated the **Figure 4** to a better tokenization. We also refer **R-mB11** to the experiments of appendix D.4 which assesses the impact of the different quantizer losses on the resulting tokenization.

---

> > ### Author Response · Authors · 2024-12-02
> > **Answer to Reviewer R-mB11 (2/2)**
> >
> > # 5. About the use of a transformer
> >
> > > W5. The reason for using a transformer for sequence generation is unclear and thorough reasoning for using it would be appreciated as compared to HIQL.
> >
> > > Q3. Elaborate on the reasons for using a transformer architecture in and ablations for the same would be helpful.
> >
> > Our use of a transformer comes from the need of planning in distribution high-level trajectories. In offline RL, the shortest path does not guarantee a better success rate because if another path is more in-distribution with the dataset, this other path could lead to better low policy success. Token prediction, as we do, discourages paths where data is scarce. Because of this, we chose a transformer which acts as a temporally coherent sequence generator that allows for trajectory generation that is inline with the dataset.
> >
> > # 6. About performances on the smaller scale environments
> >
> > > W6. Does not perform well on smaller antmaze examples compared to other methods like HIQL.
> >
> > > Q4. Why does QPHIL not perform above SOTA in smaller Antmaze datasets? Could changing the way of tokenization help with this?
> >
> > Given that this comment was raised by multiple reviewers, we kindly refer **R-mB11** to the our **“Answer to all reviewers”** in the **“About performances on the smaller scale environments”** section, where we detail why we believe that the experiments shows that QPHIL’s is of great interest for solving hard navigation problems and stays competitive for smaller environments while scaling much better than the current approaches.
> >
> > # 7. Minor Nitpicks
> > We thank the reviewer for highlighting those typos. We fixed them in the revised version of the paper.
> >
> > # 8. QPHIL performance with and without data augmentation
> > > Q5. Which is the proposed method in the paper, “QPHIL w/ aug.” or “QPHIL w/o aug.”? Having both where-in “w/ aug.” performs better in some scenarios and “w/o aug.” otherwise is confusing and elaborating on that would be helpful.
> >
> > We thank **R-mB11** for highlighting the need for clarification on this point. Since this concern is also raised by **R-hhSM**, we kindly refer you to the** "Answer to All Reviewers"** section, specifically the **"QPHIL performance with and without data augmentation"** subsection. In this section, we provide a detailed explanation of the reasons behind the observed performance differences and propose an additional experiment to further illustrate our findings.
> >
> > # 9. How can QPHIL be applied on other navigation tasks?
> >  > Q6. Can the prescribed method QPHIL work on other navigation tasks like Autonomous driving, and how would tokenization work for such a task?
> >
> > While we did not consider the application of QPHIL in autonomous driving in this work, we agree with the reviewer on the fact that it could be an interesting use case. In the following we share our insights about the application of QPHIL to autonomous driving. As QPHIL is meant to solve long term navigation problems, the autonomous driving setting seems to be a very good application of QPHIL, which could leverage a high quantity of demonstration data to learn offline without risking crashes with online interactions. Also, while methods such as HIQL would struggle to tackle those very long range navigation settings as they aim to learn a unique value function for both hierarchy levels, QPHIL tackles the issue of the signal-to-noise ratio by dissociating path planning and path following through the discretization of the navigation space. Given a high level information like GPS position, and also low level information such as camera or lidar information, a quantizer could be applied to GPS information to discretize the map of the road to perform a navigation plan, while the low level policy could leverage all low level information to perform high quality driving, without the risk of being trained with a noisy value function. Indeed, lowering the noise of generated subgoals would also seem important to reduce the risk of accidents.

---

### Official Review · Reviewer_hhSM · 2024-11-04

**Soundness:** 2
**Presentation:** 3
**Contribution:** 1
**Rating:** 3
**Confidence:** 3

**Summary:**

This paper proposed a hierarchical offline RL framework, QPHIL, for navigation tasks. It relies on discretizing the state space into landmarks by learning a latent state representation and quantization. The proposed QPHIL algorithm then trains a high-level landmark planning policy via behavior cloning and a low-level goal-conditioned policy via IQL. QPHIL achieved good performance on the D4RL Antmaze benchmark and outperformed existing flat and hierarchical offline RL and BC baselines.

**Strengths:**

The idea of quantifying the states into coarse landmarks for high-level planning in hierarchical RL is interesting and intuitive. The proposed algorithm outperformed existing approaches by a large margin on the AntMaze benchmark.

**Weaknesses:**

1. Although multiple variants of the AntMaze environment were considered, the algorithm was only tested in the AntMaze environment. It should be tested for additional and possibly more complex navigation tasks (e.g., visual navigation, autonomous driving). One crucial factor is to show that the proposed tokenization method can scale to high-dimensional observation space (e.g., images).
2. I also wonder if learning a latent space and quantizing it in the tested AntMaze environment is necessary. According to the visualized examples, the learned landmarks are mostly uniformly distributed across the maze and similar in size. Can one simply discrete the maze into lattices without learning the latent space? It also links back to the first point. The experiments would be more convincing if additional navigation tasks with high-dimensional visual observations could be included.

**Questions:**

1. In some of the tasks in Table 1, introducing data augmentation hindered QPHIL's performance by a large margin. Could the authors explain why data augmentation was not effective in these cases? It makes sense to me that data augmentation does not necessarily improve performance. Still, it is counterintuitive that the data augmentation step could cause a significant drop in performance in this context.
2. Figure 3 is a bit confusing. How is $\omega$ defined, and why is the subgoal generation terminated once $\omega$ is reached?
3. The signal-to-noise ratio arguments in Sec. 3 are hard to follow. In particular, I would appreciate the authors elaborating on the points: "A high k would improve the high policy’s signal-to-noise ratio by querying more diverse subgoals but at the cost of decreasing the signal-to-noise ratio of the low policy. Conversely, a low k would improve the low policy’s signal-to-noise ratio by querying values for nearby goals but at the cost of the diversity of the high subgoals."

---

> ### Author Response · Authors · 2024-12-02
> **Answer to Reviewer R-hhSM (1/2)**
>
> We thank Reviewer **R-hhSM** for their insightful feedback and for acknowledging the strengths of our approach. We are pleased that you found the idea of quantifying states into coarse landmarks for high-level planning in hierarchical reinforcement learning (RL) to be both **interesting** and **intuitive**. Additionally, we appreciate your recognition of our algorithm's **significant performance improvements on the AntMaze benchmark**. Your comments provide valuable validation of our methodology, and we have taken them into consideration to further refine and enhance our work.
>
> # 1. On the selection of evaluation benchmarks
> > W.1 Although multiple variants of the AntMaze environment were considered, the algorithm was only tested in the AntMaze environment. It should be tested for additional and possibly more complex navigation tasks (e.g., visual navigation, autonomous driving). One crucial factor is to show that the proposed tokenization method can scale to high-dimensional observation space (e.g., images).
>
> We thank **R-hhSM** for raising this concern. Since other reviewers have also highlighted this issue, we kindly refer you to the **"Answer to All Reviewers"** section, specifically the **"On the selection of evaluation benchmarks"** subsection, where we provide a comprehensive response.
>
> # 2. On the value of the learning of the latent space
> > W.2 I also wonder if learning a latent space and quantizing it in the tested AntMaze environment is necessary. According to the visualized examples, the learned landmarks are mostly uniformly distributed across the maze and similar in size. Can one simply discrete the maze into lattices without learning the latent space? It also links back to the first point. The experiments would be more convincing if additional navigation tasks with high-dimensional visual observations could be included.
>
> In the tested Antmaze environment, it is indeed possible to discretize uniformly the maze to perform planning. However, this approach would be limited because of several points. First, it requires the user to have previous knowledge about the shape of the maze, to choose small enough lattices to avoid landmarks to span across obstacles, which is not necessarily the case and is a strong hypothesis that our method doesn't require. Also, the size of those lattices could become really small for some environments, leading to an unnecessary increase of the number of tokens, resulting in more difficult planning. Additionally, we ran experiments with uniform tokenization on the antmaze-ultra-diverse-v0 dataset with a comparable amount of tokens than what we used for the main experiments. This resulted in a score of 53.5 ± 12.7, which is 10 to 20 pourcent lower than with the quantizer method. We reported the figure of the resulting tokenization as well as our explanation in **appendix D.7**.
>
> # 3. QPHIL performance with and without data augmentation
> > Q.1 In some of the tasks in Table 1, introducing data augmentation hindered QPHIL's performance by a large margin. Could the authors explain why data augmentation was not effective in these cases? It makes sense to me that data augmentation does not necessarily improve performance. Still, it is counterintuitive that the data augmentation step could cause a significant drop in performance in this context.
>
> We thank the reviewer **R-hhSM** for pointing out this need for clarification. As this question is also raised by R-mB11, please refer to the **"Answer to All Reviewers"** section, particularly the **"QPHIL performance with and without data augmentation"** subsection, where explain the reason behind such differences in performances as well as proposing an additional experiment to illustrate our statement.

---

> > ### Author Response · Authors · 2024-12-02
> > **Answer to Reviewer R-hhSM (2/2)**
> >
> > # 4. Questions about Figure 3
> > > Q.2 Figure 3 is a bit confusing. How is omega defined, and why is the subgoal generation terminated once is omega reached?
> >
> > **Figure 3** describes the open-loop inference of our model. omega is a given token of the generated plan. Our inference pipeline, which follows the text of section 4.1 with notations defined in it, and also the pseudocode of algorithm 1 given in appendix B, can be decomposed in two parts: path planning and path following.
> >
> > **Path planning:** The environment is initialized by generating a goal g to reach along with a starting state s0 (grey box). Then, the quantizer encodes g and s0 into the tokens of their respective landmarks (red box). The transformer uses those tokens to generate a path of landmark tokens that goes from the starting landmark to the goal landmark, called the subgoal sequence.
> >
> > **Path following:** In parallel, the current state (s0 at start and then sk) is tokenized and compared to the goal token. If it is the same, we call the goal reaching policy to precisely reach the goal. Otherwise, this current state token is compared to the subgoal token omega. If it is the same, we discard omega and take the next subgoal token from the plan until we have a new omega. With this new subgoal token omega, we ask the subgoal policy to give an action to reach the landmark of omega, which leads to a new state, and we repeat this process until the goal is reached.
> >
> > # 5. About the signal-to-noise argument
> >
> > > Q.3 The signal-to-noise ratio arguments in Sec. 3 are hard to follow. In particular, I would appreciate the authors elaborating on the points: "A high k would improve the high policy’s signal-to-noise ratio by querying more diverse subgoals but at the cost of decreasing the signal-to-noise ratio of the low policy. Conversely, a low k would improve the low policy’s signal-to-noise ratio by querying values for nearby goals but at the cost of the diversity of the high subgoals."
> >
> > We thank **R-hhSM** for highlighting the need for clarification.This point comes from section 4 of [1]. In offline RL, the objective is to learn a policy $\pi(a | s_t, g)$ that seeks to maximize a learned goal-conditioned value function $V(s_{t+1}, g)$. In sparse rewards settings, the reward signals regarding actions are delayed until the reaching of the goal, which can lead to the learning of a noisy value function. For far away goals, the differences between the estimated values of the next states $V(s_{t+1}, g)$ can be small, as bad actions can be corrected in the future without presenting a high cost on the accumulated rewards. Also, the estimated value for faraway goals is increasingly noisy and can overshadow the actual signal value. HIQL proposes to use a hierarchical framework to learn two policies: $π(a_t | s_t, s_{t+k})$ and $\pi(s_{t+k} | s_t, g)$ from the same value function. Learning $\pi(a_t | s_t, s_{t+k})$ through $V(s_{t+1}, s_{t+k})$ is better since $(s_{t+1}, s_{t+k})$ are closer, so the noise is reduced. Learning $\pi(s_{t+k} | s_t, g)$ through $V(s_{t+k},g)$ is better since the possible st+k to choose from are more diverse, so the signal is better. Hence, the signal-to-noise ratio is improved in both hierarchical levels. We provided in **appendix A** details about the IQL and HIQL baselines as well as more details about the aforementioned point and we kindly refer to this section of the paper.
> >
> > [1] HIQL: Offline Goal-Conditioned RL with Latent States as Actions

---

### Official Review · Reviewer_E6v9 · 2024-11-04

**Soundness:** 3
**Presentation:** 3
**Contribution:** 2
**Rating:** 6
**Confidence:** 3

**Summary:**

The paper proposes a hierarchical policy architecture to address challenges in learning long-horizon goal conditioned policy with an offline RL setup where value function estimation can be particularly noisy. To decouple long-horizon planning and low-level control, the authors propose training a high-level agent that leverages a learned discretization of the state space (landmarks) to propose plans which are then achieved by low-level policies that transfer the agent between landmarks and also within a landmark (for precise goal alignment).

**Strengths:**

* The paper is well motivated and presents the approach clearly. The experiments also support the key claim of improvements in longer-horizon tasks.
* The proposed method’s discretization scheme and explicit trajectory stitching for learning high-level planning from offline data is an interesting approach and can be applied to more settings.

**Weaknesses:**

- The paper is missing some details on the impacts of codebook size and tuning of the contrastive loss weights on the performance of the approach, these parameters seem integral to the contribution of the paper and also for broader applicability. As identified by the paper the subgoal step parameter of k is important in balancing the signal-to-noise ratio in low-level and high-level updates, as a consequence of replacing the training of high level policy with BC by a transformer it might have been beneficial to have very granular codebooks?
- While the proposed explicit trajectory stitching by augmenting high-level plans with different achievable subsequences works for the settings considered in the task it might not be broadly applicable as different paths have different difficulties (achievable success rates). So in general one might still have to learn a value function to perform implicit stitching – to isolate the benefits of proposed discretization at high-level, and also serve as a more direct comparison with HIQL, a version where a high-level value function is learned for planning can be beneficial to strengthen the claims.

**Questions:**

- Are all the results of QPHIL presented operating on open loop high level plans synthesized by the transformer policy on seeing just the first state?
- The hyperparameter for VQ-VAE suggests a very high coefficient for contrastive loss over reconstruction loss – what are suitable scales to balancing these different losses? Did they have to be tuned for different scales of the maze?
- With the stitched trajectory augmentation, I am wondering if the high-level BC policy has a tendency to generate longer paths if sampled multiple times. How many more trajectories are generated by such augmentation? Another assumption for stitching is that the low-level policy is capable of achieving any path – does the coverage of the dataset support even learning such policies?
- How does the coverage of the dataset (in the state space) broadly impact both the learning of the VQ-VAE and the policy?

---

> ### Author Response · Authors · 2024-12-02
> **Answer to Reviewer R-E6v9 (1/2)**
>
> We greatly appreciate **R-E6v9** for their encouraging feedback and for recognizing the strengths of our work. We are delighted that you found our paper well-motivated and clearly presented, and that our experimental results effectively support our claims of enhanced performance in longer-horizon tasks. Furthermore, we are thankful that our discretization scheme and explicit trajectory stitching for high-level planning from offline data were viewed as innovative approaches with broad applicability to various settings.
>
> # 1. About codebook size and continuous planning
>
> > W.1 The paper is missing some details on the impacts of codebook size and tuning of the contrastive loss weights on the performance of the approach, these parameters seem integral to the contribution of the paper and also for broader applicability. As identified by the paper the subgoal step parameter of k is important in balancing the signal-to-noise ratio in low-level and high-level updates, as a consequence of replacing the training of high level policy with BC by a transformer it might have been beneficial to have very granular codebooks?
>
> We thank the reviewer for suggesting such experiments. As this interrogation is shared with **R-XZDy**, we kindly refer to the **"Answer to all reviewers"** part, in the **"About codebook size and continuous planning"** section which deals with the impact of the codebook granularity on QPHIL’s overhaul performance.
>
> # 2. About replanning
> > Q.1 Are all the results of QPHIL presented operating on open loop high level plans synthesized by the transformer policy on seeing just the first state?
>
> We thank the reviewer **R-E6v9** for pointing out this need for clarification. As this question is also raised by **R-XZDy**, please refer to the **"Answer to All Reviewers"** section, particularly the **"About replanning"** subsection, where we explore the effect of replanning integration on QPHIL’s performance.
>
> # 3. On the impact of contrastive loss
> > Q.2 The hyperparameter for VQ-VAE suggests a very high coefficient for contrastive loss over reconstruction loss – what are suitable scales to balancing these different losses? Did they have to be tuned for different scales of the maze?
>
> As you mentioned the coefficients for the learning of the quantizer (the commit loss coefficient of the VQ-VAE, the contrastive loss coefficient as well as the reconstruction loss coefficient) need tuning. As we use the same coefficients for each loss for each maze shape, we can argue that our set of values are generalizable to every shape and size of the tested mazes. The tuning of those losses was the result of hyperparameter search. Moreover, following **R-E6v9** suggestion, we added in **appendix D.4** a more comprehensive analysis of the impact of the contribution of each loss to the learning of the space discretization.

---

> > ### Author Response · Authors · 2024-12-02
> > **Answer to Reviewer R-E6v9 (2/2)**
> >
> > # 4. About data augmentation
> > > Q.3 With the stitched trajectory augmentation, I am wondering if the high-level BC policy has a tendency to generate longer paths if sampled multiple times. How many more trajectories are generated by such augmentation? Another assumption for stitching is that the low-level policy is capable of achieving any path – does the coverage of the dataset support even learning such policies?
> >
> > We thank the reviewer for this relevant remark. A high-level BC might indeed have a small tendency to generate indeed larger paths. However, this issue is mitigated by the fact that in most cases, the most likely path to reach a goal is a shorter one. We share here the table of the number of trajectories with and without such data augmentation, also added in in **appendix D.5**:
> >
> > | Dataset                        | wo/ stitching | w/ stitching (mean ± std)         |
> > |--------------------------------|---------------|------------------------------------|
> > | antmaze-medium-diverse-v0      | 999 ± 0       | 3696.5 ± 298.14                   |
> > | antmaze-medium-play-v0         | 999 ± 0       | 4092.375 ± 354.36                 |
> > | antmaze-large-diverse-v0       | 999 ± 0       | 12106.375 ± 1421.35               |
> > | antmaze-large-play-v0          | 999 ± 0       | 12786.625 ± 2578.04               |
> > | antmaze-ultra-diverse-v0       | 999 ± 0       | 11730.875 ± 895.24                |
> > | antmaze-ultra-play-v0          | 999 ± 0       | 11687.75 ± 1280.93                |
> > | antmaze-extreme-play-v0        | 499 ± 0       | 10203.125 ± 628.96                |
> > | antmaze-extreme-diverse-v0     | 499 ± 0       | 9400.25 ± 269.02                  |
> >
> > Regarding the stitching, the fact that the low-level can achieve any paths can be supported by the contrastive loss of the tokenization that leads the quantization to form single piece navigable landmarks. As such, navigating from a landmark to another is possible as long as there exists a sufficient number of transitions that perform such navigation in the demonstration data. Furthermore, as the transformer predicts likely trajectories, it will select paths of high likelihood where the low policy is more likely to perform because of a high demonstration data density. Also, while we acknowledge the diversity of data augmentation techniques available, we focused our approach on this mechanism which provided the best results across the compared methods.
> >
> > # 5. On the impact of the dataset coverage
> >
> > > Q.4 How does the coverage of the dataset (in the state space) broadly impact both the learning of the VQ-VAE and the policy ?
> >
> > We see experimentally in **appendix D.6** that areas with very low to null coverage (specifically walls here) share the same token as one of the nearest in-distribution states, showing a certain amount of generalization of the tokenization. Also, high coverage areas have a tendency to constrain a higher number of smaller landmarks than the low coverage parts of the maze, due to the loss having more weight in those, since there are more samples. **Section 5.4** illustrates that this issue is mitigated by the contrastive loss, leading to tokens of more uniform size. For the policy, as we use an offline reinforcement learning (IQL), low coverage areas are to be avoided and IQL seeks to sample actions within training distribution to avoid getting out-of-distribution.

---

### Official Review · Reviewer_XZDy · 2024-11-04

**Soundness:** 2
**Presentation:** 3
**Contribution:** 2
**Rating:** 5
**Confidence:** 4

**Summary:**

The paper identifies a key challenge in offline RL: the signal-to-noise ratio, noting that prior work has shown the benefits of hierarchical offline RL methods in addressing this issue. In response, the authors introduce a hierarchical, transformer-based approach for goal-conditioned offline RL. Their method learns landmarks through a VQ-VAE model and plans a sequence of subgoals using a transformer-based planner, with subgoal policies trained via IQL. To encourage meaningful temporal structure, the authors implement a contrastive loss that assigns the same tokens to temporally close states while differentiating distant states. Additionally, they propose a subgoal-level stitching mechanism as a form of data augmentation. The method is evaluated on long-horizon tasks such as AntMaze, benchmarking against HIQL and other baselines. The experiments also include an ablation study to analyze the contributions of each component to performance improvements.

**Strengths:**

1. The proposed methods demonstrated significant improvement in the AntMaze environment, particularly on the Ultra maze, while achieving comparable results on smaller mazes.
2. The authors introduce a contrastive loss function that encourages temporally close states to share the same tokens, while assigning different tokens to temporally distant states. This contrastive loss yields substantial performance gains, particularly in the AntMaze-Extreme setting.
3. Leveraging the tokenization of trajectories, the authors propose a subgoal-level stitching mechanism as a form of data augmentation, which enhances performance in most cases.

**Weaknesses:**

1. A primary concern with the results is that they are limited to the navigation domain, specifically the AntMaze environment. This raises questions about the generalizability of the method to other settings, such as the Kitchen or Calvin environments evaluated in the HIQL paper. Expanding the evaluation to include diverse tasks would provide a clearer understanding of the method's broader applicability.
2. The concept of discrete planning over learned landmarks, subgoals, or skills has been well-explored in both online RL (e.g., Choreographer [1], Dr. Strategy [2], CQM [3]) and offline RL (e.g., PTGM [4], SAQ [5], SkillDiffuser [6], TAP [7]). While I acknowledge that none of these prior works directly address the goal-conditioned offline RL setting and that they are slightly methodologically different, the novelty here appears limited and incremental.
3. The authors state, “For long-distance tasks, the signal-to-noise ratio still degrades during subgoal generation, which can result in a noisy high-level policy and, consequently, reduced performance. In this paper, we propose to shift the learning paradigm of the high-policy towards discrete space planning” (lines 56-61). However, their analysis lacks a direct comparison between discrete and continuous space planning within the same framework to validate its impact on mitigating this issue. I would suggest modifying their approach by training a VAE instead of a VQ-VAE to enable such a comparison as part of an ablation study. Alternatively, they could compare their method to a variant using a larger number of landmarks to assess how this affects performance in handling the signal-to-noise problem.

[1] Mazzaglia, Pietro, Tim Verbelen, and Bart Dhoedt. "Choreographer: learning and adapting skills in imagination." ICLR, the 11th International Conference on Learning Representations. 2023.

[2] Hamed, Hany, et al. "Dr. Strategy: Model-Based Generalist Agents with Strategic Dreaming." Forty-first International Conference on Machine Learning.

[3] Lee, Seungjae, et al. "Cqm: Curriculum reinforcement learning with a quantized world model." Advances in Neural Information Processing Systems 36 (2023): 78824-78845.

[4] Yuan, Haoqi, et al. "Pre-training goal-based models for sample-efficient reinforcement learning." *The Twelfth International Conference on Learning Representations*. 2024.

[5] Luo, Jianlan, et al. "Action-quantized offline reinforcement learning for robotic skill learning." *Conference on Robot Learning*. PMLR, 2023.

[6] Liang, Zhixuan, et al. "Skilldiffuser: Interpretable hierarchical planning via skill abstractions in diffusion-based task execution." *Proceedings of the IEEE/CVF Conference on Computer Vision and Pattern Recognition*. 2024.

[7] Zhang, Tianjun, et al. "Efficient Planning in a Compact Latent Action Space." The Eleventh International Conference on Learning Representations.

**Minor Improvements (Not considered in the score)**

Some of the above-mentioned papers are not cited in the current draft. Including these in the citations and discussing their distinctions in the related work section would provide a more comprehensive view of prior approaches. Highlighting these works would clarify the differences in how they approach discrete planning of learned landmarks, subgoals, or skills and help position the contribution of this paper within the broader context of online and offline RL. Specifically, emphasizing the unique aspects of goal-conditioned offline RL in comparison to these methods would strengthen the novelty argument.

**Questions:**

1. Could you provide more details on how you selected the number of landmarks? Additionally, how does varying the number of landmarks impact performance? It would be helpful to understand how different quantities affect the system’s effectiveness.
2. In the experiments, could you clarify the terms “w/ repr.” and “w/o repr.”? I couldn’t find an explanation in the text.
3. From Figure 3, it appears that planning occurs only once. How does the system handle situations where the agent deviates from the plan? Is there any mechanism for re-planning if the agent is unable to follow the initial trajectory?

---

> ### Author Response · Authors · 2024-12-02
> **Answer to Reviewer R-XZDy**
>
> We are thankful to **R-XZDy** for their review. We are thankful that our work is acknowledged for its significant improvement over the previous methods on difficult long range navigation tasks, thanks to our choice of a contrastive loss function that allows an adaptive discretization of the state space, which can be leveraged for planning thanks to data augmentation through stitching.
>
> # 1. On the selection of evaluation benchmarks
>
> > W.1 A primary concern with the results is that they are limited to the navigation domain, specifically the AntMaze environment. This raises questions about the generalizability of the method to other settings, such as the Kitchen or Calvin environments evaluated in the HIQL paper. Expanding the evaluation to include diverse tasks would provide a clearer understanding of the method's broader applicability.
>
> We appreciate **R-XZDy** for bringing up this concern. As this question has been raised by multiple reviewers, we kindly direct you to the **"Answer to All Reviewers"** section, particularly the **"On the Selection of evaluation benchmarks"** subsection, where we address this topic in detail.
>
> # 2. About additional related works
>
> > W.2 The concept of discrete planning over learned landmarks, subgoals, or skills has been well-explored in both online RL (e.g., Choreographer [1], Dr. Strategy [2], CQM [3]) and offline RL (e.g., PTGM [4], SAQ [5], SkillDiffuser [6], TAP [7]). While I acknowledge that none of these prior works directly address the goal-conditioned offline RL setting and that they are slightly methodologically different, the novelty here appears limited and incremental.
>
> We sincerely thank **R-XZDy** for their insightful feedback and for providing precise references to support their concerns regarding the novelty of our work. We appreciate the opportunity to clarify our contributions and address these points in detail. As we consider that this question might be of interest for all reviewers, we address it in the  **"Answer to All Reviewers"** section in the **"About additional related works"**.
>
> # 3. About codebook size and continuous planning
>
> > W.3 I would suggest modifying their approach by training a VAE instead of a VQ-VAE to enable such a comparison as part of an ablation study. Alternatively, they could compare their method to a variant using a larger number of landmarks to assess how this affects performance in handling the signal-to-noise problem.
>
> > Q.1 Could you provide more details on how you selected the number of landmarks? Additionally, how does varying the number of landmarks impact performance? It would be helpful to understand how different quantities affect the system’s effectiveness.
>
> We appreciate **R-XZDy** for recommending ablation experiments to compare continuous and discrete state planning. Since **R-E6v9** has also raised this point, please refer to the **"Answer to All Reviewers"** section, particularly the **"About codebook size and continuous planning"** subsection, where we provide a comprehensive response.
>
> # 4. About  “w/ repr.” and “w/o repr.”
> > Q.2 In the experiments, could you clarify the terms “w/ repr.” and “w/o repr.”? I couldn’t find an explanation in the text.
>
> We thank the reviewer for pointing out this omission. “w/ repr.” and “w/o repr.” refers to the variants of the HIQL algorithm described in [1]. HIQL “w/o repr.” uses two policies, a high level policy $\pi_{\theta_h}^h(s_{t+k}|s_t,g)$ that generates a subgoal to reach given the current state $s_t$ and the goal $g$ and $\pi_{\theta_l}^l(a_t|s_t,s_{t+k})$ that generates an action to take to reach this subgoal given $s_t$ and $s_{t+k}$. Because HIQL may handle complex high-dimensional observation spaces, generating such subgoals in raw format might be challenging. Hence, they propose to learn a representation $\phi(s)$ of the states, leading to a high level policy $\pi_{\theta_h}^h(\phi(s_{t+k})|s_t,g)$ and the low level policy $\pi_{\theta_l}^l(a_t|s_t,\phi(s_{t+k}))$, leading to the “w/ repr.” variant. We display the two in our tables for a better comparison. **This is now clearly stated in the paper below Table 1.**
>
> [1] HIQL: Offline Goal-Conditioned RL with Latent States as Actions
>
> # 5. About replanning
> > Q.3 From Figure 3, it appears that planning occurs only once. How does the system handle situations where the agent deviates from the plan? Is there any mechanism for re-planning if the agent is unable to follow the initial trajectory?
>
> We thank the reviewer for pointing this out and kindly refer to the **“Answer to all reviewers”** part, in the section **“About replanning”**, which delves into the impact of replaning integrations to our method.

---

> > ### Comment · Reviewer_XZDy · 2024-12-03
> > **Official Comment by Reviewer XZDy**
> >
> > It is extremely hard to go through all the responses thoroughly as the responses are submitted very late just before our deadline to discuss with the authors.
> >
> > However, I would like to thank the authors for their replies. I will try to reply as much as I can due to the time constraints.
> >
> > Regarding "**4. About “w/ repr.” and “w/o repr.”**":
> > I see, thank you very much for the elaboration. That is indeed was not clear from the previous description.
> >
> > I will reply to the rest in the "**Answer to All Reviewers**"

---

> > > ### Comment · Reviewer_XZDy · 2024-12-03
> > > **Official Comment by Reviewer XZDy**
> > >
> > > After carefully reading all the replies, I will keep my score "5: marginally below the acceptance threshold."
> > >
> > > **Why?**
> > >
> > > I fully consider the provided contributions in this paper; I think the provided method is effective enough in the tested settings, the provided analysis is very useful, and the provided solutions like contrastive loss and augmentation are novel.
> > >
> > > But, I believe that you still need to provide other results for different environments, as I mentioned in the common response section.

---

### Official Review · Reviewer_mPVU · 2024-11-07

**Soundness:** 2
**Presentation:** 3
**Contribution:** 2
**Rating:** 5
**Confidence:** 4

**Summary:**

This paper focuses on hierarchical methods in offline goal-conditioned reinforcement learning (GCRL) and proposes an optimized hierarchical framework specifically for this domain. The paper makes contributions by introducing some structures like VQ-VAE to enhance the existing hierarchical structures used in offline GCRL.

**Strengths:**

1. The focus on offline GCRL is valuable, and introducing a hierarchical framework with further in-depth exploration is interesting.
2. The writing is clear, and the paper is well-structured.
3. The paper introduces multiple interesting large-scale maze environments, which require high-quality GCRL policies to solve.

**Weaknesses:**

1. The proposed method is tested exclusively on the AntMaze benchmark, which raises concerns about the method's generalizability and robustness across different offline GCRL scenarios.
2. In many AntMaze settings, the proposed approach does not consistently outperform existing state-of-the-art methods, such as HIQL, thus limiting the evidence for its effectiveness over current approaches.

**Questions:**

The authors are encouraged to discuss the potential of applying the proposed approach to other offline GCRL environments to validate its broader applicability. Achieving successful results on varied benchmarks would strengthen the method's impact and demonstrate its generalizability.

---

> ### Comment · Reviewer_mPVU · 2024-11-25
>
> The authors didn't provide rebuttals. So I suggest rejecting this paper.

---

> > ### Author Response · Authors · 2024-11-25
> >
> > Sorry for the delay, we are currently working on the answers. We know that it would have been better to keep more time for discussion, but there are still two days remaining for giving insights as response to your valuable comments. We make our best for posting them as soon as possible.

---

> > > ### Author Response · Authors · 2024-12-02
> > > **Answer to Reviewer R-mPVU**
> > >
> > > We thank **R-mPVU** for their review. We appreciate that the chosen focus of our paper’s contribution towards hierarchical methods for GCRL was considered interesting and valuable and that **R-mPVU** found our paper clear and well-written.
> > >
> > > # 1. On the selection of evaluation benchmarks
> > > > W.1 The proposed method is tested exclusively on the AntMaze benchmark, which raises concerns about the method's generalizability and robustness across different offline GCRL scenarios.
> > >
> > > > Q.1 The authors are encouraged to discuss the potential of applying the proposed approach to other offline GCRL environments to validate its broader applicability. Achieving successful results on varied benchmarks would strengthen the method's impact and demonstrate its generalizability
> > >
> > > We thank **R-mPVU** for raising this concern. Since this issue has also been highlighted by other reviewers, we kindly refer you to the **"Answer to All Reviewers"** section, specifically the **"On the selection of evaluation benchmarks"** subsection, where we provide a comprehensive response.
> > >
> > > # 2. About performances on the smaller scale environments
> > >
> > > > W.2 In many AntMaze settings, the proposed approach does not consistently outperform existing state-of-the-art methods, such as HIQL, thus limiting the evidence for its effectiveness over current approaches.
> > >
> > > Given that this comment was raised by multiple reviewers, we kindly refer **R-mPVU** to the our **"Answer to all reviewers"** in the **"About performances on the smaller scale environments"** section, where we detail why we believe that the experiments shows that QPHIL’s is of great interest for solving hard navigation problems and stays competitive for smaller environments while scaling much better than the current approaches.

---

> > > > ### Comment · Reviewer_mPVU · 2024-12-03
> > > >
> > > > I appreciate for your replies. I indeed understand the effectiveness of the proposed method on AntMaze navigation environment and admire your contribution on constructing larger and more complicated environments. However, the proposed method  seems only be applicable on the AntMaze or Maze navigation problems unlike baseline method HIQL. What I hope is to check whether the method can be used on other domains such as Kitchen in D4RL. So I lean to keep my score at borderline reject.

---

### Author Response · Authors · 2024-12-02
**Answer to All Reviewers (1/8)**

First of all, we are deeply grateful to all reviewers for their patience as well as comprehensive and insightful comments on our manuscript.

We were pleased to see that the reviewers acknowledged the motivation of our work which focuses on the challenging and important offline Goal-Conditioned RL (GCRL) problem studied using multiple large-scale environments (**R-mPVU** and **R-E6v9**). We were also pleased to know that the reviewers appreciated the innovative temporal organization of the space into landmarks through a contrastive loss (**R-XZDy**, **R-E6v9**, **R-hhSM**, **R-mB11**), the significant improvement in performance over state-of-the-art of our approach for long-range goal conditioned navigation tasks (**R-XZDy**, **R-E6v9**, **R-hhSM**, **R-mB11**), as well as the addition to a new very long-range environment (**R-mB11**).

Following your different suggestions, we address in the following global answer the several points and questions shared by the different reviewers, while tackling specific questions to each reviewer in the dedicated answers. We also uploaded a revised version of the paper by coloring in **red** the several additions we made.

# 1. On the selection of evaluation benchmarks

We appreciate the reviewers' concerns regarding the generalizability and robustness of our proposed method, particularly in relation to its evaluation exclusively on the AntMaze benchmark. We would like to provide additional context and justification to address these points.
QPHIL is a method aimed at improving the SOTA regarding hard long-range navigation purposes, where an agent has to perform **long-term planning** as well as **solving hard locomotion tasks**. AntMaze is widely recognized within the reinforcement learning (RL) community as a **standard benchmark** for evaluating goal-conditioned and hierarchical RL methods. Its inherent complexity arises from the necessity for both **long-range high-level planning** and **non-trivial low-level locomotion**, making it an ideal testbed for assessing the capabilities of advanced RL algorithms. To ensure a thorough evaluation of our method's **generalizability and robustness**, we extended our experiments beyond the standard AntMaze environment by incorporating additional variants:

 - **AntMaze Ultra (From  [1]):** This variant introduces increased complexity compared to the standard AntMaze, presenting more intricate navigational challenges and diverse obstacle configurations. AntMaze Ultra serves as an intermediate benchmark, bridging the gap between the standard and our most challenging variant.
 - **AntMaze Extreme (Our Contribution):** AntMaze Extreme represents the most demanding variant within our evaluation suite. It incorporates extreme navigation scenarios in the form of highly complex maze structures, pushing the boundaries of existing benchmarks. This variant is designed to rigorously test the limits of hierarchical planning and low-level control mechanisms for very long-range navigation, ensuring that our method demonstrates high performance under the most challenging conditions.
 - **Random AntMaze Ultra and Random AntMaze Extreme (Our Contribution):** Those environments cycles through a diverse set of 50 couples of $(s_0, g)$ allowing a more rigorous test of the generalization capabilities of our model to diverse set of paths, which answers the questions raised about the potential variances on the results with and without data augmentation raised by the reviewers. The results on those environments are written in **section 5.5**.
While evaluating on a broader range of environments could further validate generalizability, we think that the AntMaze suite—including standard, Ultra, Extreme and Random variants—provides a first rigorous and diverse testing ground for QPHIL’s navigation scope, as well as being challenging even for state-of-the-art methods like HIQL.

[1] Efficient Planning in a Compact Latent Action Space

---

> ### Author Response · Authors · 2024-12-02
> **Answer to All Reviewers (2/8)**
>
> # 2. About codebook size and continuous planning
>
> We thank **R-XZDy** and **R-E6v9** for suggesting an analysis of the granularity of the codebook, first of all by comparing the differences between the continuous (or infinite subgoal number) and discrete subgoal paradigms, but also by analyzing the impact of different codebook sizes.
>
> First, we followed **R-XZDy**'s suggestion on analyzing the impact of shifting from discrete to continuous space planning. As our architecture corresponds to a hierarchical planning involving high level subgoal planning through imitation learning and low level subgoal reaching in a reinforcement learning framework, a simple approach for comparing both continuous and discrete subgoals would be to consider an ablated HIQL which uses behavioral cloning rather than offline reinforcement learning to learn a high-level policy. We call this approach BC+IQL and tested its performance on the antmaze-ultra datasets. In the corresponding table below, we observe similar results to HIQL, while QPHIL demonstrates better performance, highlighting the benefits of using discrete planning for longer range navigation scenarios.
>
> | Dataset        | BC+IQL w/ repr. | BC+IQL w/o repr. | QPHIL w/ aug. | QPHIL w/o aug. |
> |----------------|-----------------|------------------|---------------|----------------|
> | ultra play     | 47.4 ± 15     | 43.2 ± 19       | 64.5 ± 6.8    | 61.5 ± 6.2     |
> | ultra diverse  | 50.8 ± 11     | 51.4 ± 17       | 61.8 ± 3.7    | 70.3 ± 6.9     |
>
> We reported those **new results** in **appendix D.2** of the revised version of the pdf. We precise that the “w/ repr.” and “w/o repr.” correspond to the addition of representation learning, while “w/ aug.” and “w/o aug” correspond to the presence of data augmentation by  trajectory stitching.
> Additionally, as suggested by the reviewers **R-XZDy** and **R-E6v9**, we launched a **new experiment** to assess the impact of the codebook size on the overhaul performance on antmaze-ultra-diverse-v0, obtaining the following table, also in **appendix D.2**:
>
> | codebook_size | number of used tokens | performance |
> |---------------|------------------------|-------------|
> | 1             | 1 / 100% use          | 0           |
> | 8             | 8 / 100% use          | 0.06        |
> | 24            | 24 / 100% use         | 0.52        |
> | 48            | 48 / 100% use         | 0.861       |
> | 96            | 96 / 100% use         | 0.72        |
> | 256           | 57 / 22% use          | 0.88        |
> | 1024          | 95 / 9% use           | 0.88        |
> | 2048          | 146 / 7% use          | 0.82        |
>
> The first column corresponds to the number of available codebook vectors of the VQ-VAE quantizer. The second column corresponds to the number of used codebook vectors. We consider that a codebook vector is used if there exists a state from training trajectories that projects its encoding on it and that a landmark is formed by the set of states whose encodings are projected on the same codebook vector. We observe that after a given threshold on the number of available codebook vectors, their use percentage decreases, i.e. the quantizer does not benefit from additional codebook vectors to achieve a good quantization. The performance of the method appears to saturate after the threshold of 48 vectors. Regarding the antmaze-ultra map, this codebook size corresponds to a good trade-off ensuring an accurate “signal-to-noise'' ratio while stabilizing high-level commands.

---

> ### Author Response · Authors · 2024-12-02
> **Answer to All Reviewers (3/8)**
>
> # 3. About replanning
>
> We thank reviewers **R-XZDy** and **R-E6v9** for voicing their interrogation about the open-loop aspect of our planning. In our main evaluations, the planning occurs only once. The policy is consequently achieving subgoal landmarks sequentially, following the initially generated plan. Aware of this, we also tested a replaning strategy which consists in replaning from the current landmark to the goal when an out-of-path landmark is detected. This replanning is done by sampling several paths and choosing the shortest one. We tested this approach in the antmaze-extreme environment and we displayed the impact of this replaning in **appendix D.3** though the following table:
>
> | Dataset          | QPHIL 1-sample | QPHIL 10-samples | QPHIL no replanning |
> |------------------|----------------|-------------------|----------------------|
> | extreme play     | 35.5 ± 7.8     | 44.3 ± 16        | 50 ± 6.9            |
> | extreme diverse  | 38.5 ± 8.9     | 50.3 ± 9.4       | 39.5 ± 13           |
>
> While we observed no significant improvement in the tested maze from such replanning in the extreme play setting, our method supports replanning easily which emphasizes the great potential of the approach.

---

> > ### Comment · Reviewer_XZDy · 2024-12-03
> > **Official Comment by Reviewer XZDy for [3. About replanning]**
> >
> > Thank you for providing this result.
> >
> > It is interesting to me why **QPHIL no replanning** is better in extreme play, do you have any clue why?

---

> > > ### Author Response · Authors · 2024-12-04
> > >
> > > Thank you for raising this point. We believe that since in our setting replanning is not necessary, it can sometimes lead to longer trajectories which can slightly decrease performance. Nevertheless, the difference in performance is not significant when the replanned trajectory is chosen from a sufficient number of sampled plans (here 10-samples).

---

> ### Author Response · Authors · 2024-12-02
> **Answer to All Reviewers (4/8)**
>
> # 4. About performances on the smaller scale environments (1/2)
>
> We thank reviewers **R-mPVU** and **R-mB11** for raising the point that QPHIL does not outperform the SOTA in the smaller environments.
>
> > **R-mPVU**: "In many AntMaze settings, the proposed approach does not consistently outperform existing state-of-the-art methods, such as HIQL, thus limiting the evidence for its effectiveness over current approaches."
>
> While we agree that QPHIL does not always statistically outperform previous methods on the smaller scale maps, it always remains competitive. On small environments (medium to large mazes), QPHIL obtains comparable results than HIQL and none of the approaches appears to consistently outperform the other in the results provided in the following table and in **section 5.2**.
>
> | Dataset          | HIQL w/ repr. | HIQL w/o repr. | QPHIL w/ aug. | QPHIL w/o aug. |
> |------------------|---------------|----------------|---------------|----------------|
> | medium-play      | 84.1 ± 11     | 87.0 ± 8.4     | **92.0 ± 3.9**| 86.8 ± 3.6     |
> | medium-diverse   | 86.8 ± 4.6    | 89.9 ± 3.5     | **90.8 ± 1.8**| 87.8 ± 2.3     |
> | large-play       | 86.1 ± 7.5    | 81.2 ± 6.6     | 82.25 ± 6.4   | **88.0 ± 5.5** |
> | large-diverse    | **88.2 ± 5.3**| 87.3 ± 3.7     | 80.25 ± 3.3   | 80.5 ± 7.4     |
> | ultra-play       | 39.2 ± 15     | 56.0 ± 12      | **64.5 ± 6.8**| 61.5 ± 6.2     |
> | ultra-diverse    | 52.9 ± 17     | 52.6 ± 8.7     | 61.8 ± 3.7    | **70.3 ± 6.9** |
> | extreme-play     | 14.2 ± 7.4    | 22.8 ± 7.9     | **50.0 ± 6.9**| 40.5 ± 9.4     |
> | extreme-diverse  | 13.7 ± 5.6    | 21.9 ± 6.6     | **39.5 ± 13** | 12.5 ± 8.5     |
>
> Indeed, in the medium-maze, both methods display similar performances, with a slight advantage to QPHIL, while HIQL performs better on large-diverse slightly better than QPHIL, but also better than in the medium-maze setting. However, on ultra and extreme mazes, QPHIL clearly overperforms HIQL and the other methods. As such, while staying competitive in short-distance settings, the proposed approach of tokenization appears strongly beneficial for longer-range scenarios which are likely closer than possible real world applications. We also provide the corresponding performance curves in the following website: https://sites.google.com/view/qphil/home.
>
> Alternatively, we conducted a Welch-Student test to check the hypothesis that “HIQL w/ repr.”’s mean is less than “QPHIL w/ aug.”’s one. We display below the results:
>
> | Dataset          | t-statistic | p-value |
> |------------------|-------------|---------|
> | medium-play      | -1.91       | 0.0444  |
> | medium-diverse   | -2.29       | 0.0237  |
> | large-play       | 1.10        | 0.8558  |
> | large-diverse    | 3.60        | 0.9981  |
> | ultra-play       | -4.34       | 0.0008  |
> | ultra-diverse    | -1.45       | 0.0938  |
> | extreme-play     | -10.01      | 0.0000  |
> | extreme-diverse  | -5.16       | 0.0003  |
>
> The statistical analysis, using one-tailed t-tests to determine if HIQL's performance is less than QPHIL's, provides compelling evidence for QPHIL's superior scaling in larger maze environments. Namely:
>  - **Medium-Sized Environments:** In medium-sized environments, both tests yield p-values below the 0.05 significance level (0.0444 and 0.0237), supporting the hypothesis that QPHIL outperforms HIQL.
>  - **Large-Sized Environments:** The results in large-sized environments do not support the hypothesis. Neither test achieves statistical significance, with p-values of 0.8558 and 0.9981, indicating that HIQL may perform comparably or better in these cases.
>  - **Ultra-Sized Environments:** In ultra-sized environments, one test achieves strong statistical significance (p = 0.0008), while the other remains insignificant (p = 0.0938). These results indicate QPHIL's emerging advantage in handling the increased complexity of these environments, despite some variability.
>  - **Extreme Environments:** The strongest evidence for QPHIL's advantage appears in the largest "extreme" environments. Both tests achieve extremely high statistical significance (p = 0.0000 and p = 0.0003), conclusively rejecting the null hypothesis. These results confirm QPHIL's superior performance in handling the challenges of extreme maze environments.
>
> **In conclusion**, The p-value analysis clearly demonstrates QPHIL's scaling advantage over HIQL, especially in ultra and extreme environments. While the differences are less significant in smaller environments, the statistical significance of the results in ultra and extreme settings supports the conclusion that QPHIL excels in handling increasingly complex tasks.

---

> ### Author Response · Authors · 2024-12-02
> **Answer to All Reviewers (5/8)**
>
> # 4. About performances on the smaller scale environments (2/2)
>
> > **R-mB11:** "Why does QPHIL not perform above SOTA in smaller Antmaze datasets? Could changing the way of tokenization help with this?"
>
> The conducted experiments such as in **section 5.2** as well as **appendix D.2** show that QPHIL benefits from the discretization of the state space, allowing to break the problem into n distinct navigation problems as **R-mB11** mentioned. We argue that those improvements come from the improvement of the signal-to-noise ratio of the learned value function, which worsens the quality of the high-level planning in methods such as HIQL which aims to learn a unique value function across the entire state space.
>
> By design, HIQL imposes a tradeoff regarding the hyperparameter controlling the distance of high-level subgoals. If this distance is high, high-level value estimates will be simpler to perform (high signal-to-noise ratio) but the low-level policy will be facing harder navigation challenges (subgoals will be distant), and vice-versa.
>
> Hence, as QPHIL learns the value function only for the low-level policy to reach the next planned landmark from its current landmark, the signal-to-noise ratio remains quite low as the long-range planning is performed with an independent planner in a discretized space. As such, for small mazes, HIQL does not suffer too much from a bad signal-to-noise ratio and consequently the two approaches display similar performances. However, for long range scenarios, HIQL’s signal to noise ratio worsens while QPHIL’s remains good, which explains why it is scalable to very long range scenarios in contrast to HIQL.

---

> ### Author Response · Authors · 2024-12-02
> **Answer to All Reviewers (6/8)**
>
> # 5. QPHIL performance with and without data augmentation
>
> We thank the **R-hhSM** and **R-mB11** for expressing their concerns. The proposed data augmentation serves to create longer trajectories than the initial ones by explicit trajectory stitching. However, those are not necessary in the antmaze-medium to antmaze-ultra datasets. Hence, data augmentation can create new unnecessary trajectories. The traditional D4RL antmaze environments used have very similar starting and goal positions for evaluation. Hence, because of the discretization, it is likely that our model converges to propose a good and unique trajectory which is sufficient to get very good performance. We created a variant of those environments called random-antmaze in **section 5.5** where the starting state and the goal are selected randomly in the maze. In this setting, the difference between augmenting or not the data is no longer significant for the ultra mazes, while being essential in the extreme mazes. This suggests that the aforementioned performance disparity is due to the poor diversity of evaluation configuration of those traditional environments and that data augmentation is neutral for smaller-range settings and beneficial for longer-range settings.

---

> ### Author Response · Authors · 2024-12-02
> **Answer to All Reviewers (7/8)**
>
> # 6. About additional related works (1/2)
>
> Thanks to **R-XZDy**’s suggestion, we added in the following a discussion about suggested additional related work to highlight the novelty of our method. While it is accurate that previous research has explored discrete planning over learned structures, our work uniquely integrates these concepts within the **goal-conditioned offline reinforcement learning (RL)** framework—a context that, as rightly pointed out by the reviewer, has not been directly addressed in prior studies.
>
> Specifically, although several cited works utilize VQ-VAEs to define landmarks or similar constructs, **none of these approaches incorporate a time contrastive** loss, the utility of which we demonstrate comprehensively in **subsection 5.4** but also in **appendix D.4**. This contrastive loss plays a crucial role in enhancing the temporal coherence of the learned representations, thereby improving the effectiveness of our hierarchical planning mechanism.
>
> Furthermore, our introduction of **explicit trajectory stitching** as a form of data augmentation is, to our knowledge, novel within the realm of offline GCRL. While alternative planning methods, such as shortest path algorithms over graphs [3], offer advantages in terms of speed and explainability, our **sequence generator approach** provides distinct benefits tailored to offline learning scenarios. Unlike shortest path methods, our approach prioritizes paths that are more aligned with the dataset, thereby **enhancing success rates by discouraging routes where data is sparse** through effective token prediction.
>
> Below, we provide a detailed comparison paper by paper to elucidate the distinct contributions of our work:
>
>  1. **Choreographer [1] (added to the related work)**
>      - **Overview:** Choreographer employs a VQ-VAE to discretize states into discrete ‘skills’. Policies are trained to reach states corresponding to their assigned skill-conditioning codes.
>      - **Comparison:** While both Choreographer and our work utilize VQ-VAEs for state discretization, our approach introduces a **time contrastive loss** that enhances temporal coherence by encouraging temporally close states to share the same tokens. This results in more robust and meaningful landmark representations, which are crucial for effective hierarchical planning in offline settings. Additionally, Choreographer focuses primarily on skill discovery without integrating explicit trajectory stitching, which our method leverages to improve data augmentation and planning robustness.
>  2. **Dr. Strategy [2] (added to the related work)**
>      - **Overview:** Dr. Strategy uses a VQ-VAE to define landmarks and conditions policies on these landmarks, primarily within an online RL framework.
>      - **Comparison:** Our method extends the concept of landmark-based planning into the **goal-conditioned offline RL** setting, addressing challenges specific to offline data constraints. Unlike Dr. Strategy, which is tailored for online interactions, our approach incorporates **explicit trajectory stitching** and **planning that aligns with data quantity**, ensuring that the planned paths are supported by ample data, thereby enhancing reliability and performance in offline environments.
>  3. **CQM [3] (Curriculum Reinforcement Learning with a Quantized World Model) (in the related work)**
>      - **Overview:** CQM employs a VQ-VAE to quantize the goal space, treats landmarks as vertices in a graph, evaluates edge costs using Q-values, and utilizes Djikstra’s algorithm for planning within an online RL framework.
>      - **Comparison:** While CQM focuses on optimizing planning through Djikstra’s algorithm in an online context, our approach is specifically designed for offline RL. We prioritize **planning paths that are well-represented within the dataset**, leveraging our contrastive loss to ensure that our sequence generator predicts paths aligned with data availability. This strategy mitigates the risk of following suboptimal paths in low-data regions, a critical consideration in offline settings where data scarcity can significantly impact performance.
>  4. **PTGM [4] (Pre-training Goal-Based Models for Sample-Efficient Reinforcement Learning)**
>      - **Overview:** PTGM utilizes simple k-means clustering for state discretization and applies this discretization after training a high-level policy in a semi-online manner.
>      - **Comparison:** Unlike PTGM’s semi-online discretization, our method employs **explicit trajectory stitching** tailored for purely offline RL scenarios. This allows us to effectively augment the dataset by stitching trajectories, enhancing the robustness of our high-level planning without relying on online interactions. Furthermore, our use of **contrastive loss** provides additional robustness by ensuring temporal coherence in the discretized state space.

---

> ### Author Response · Authors · 2024-12-02
> **Answer to All Reviewers (8/8)**
>
> # 6. About additional related works (2/2)
>
> 5. **SAQ [5] (Action-Quantized Offline Reinforcement Learning for Robotic Skill Learning) (already in the related work)**
>      - **Overview:** SAQ uses a VQ-VAE to discretize the action space but does not incorporate planning mechanisms.
>      - **Comparison:** While SAQ focuses on action space discretization, our work extends this concept by applying **discretization to the state space** and integrating comprehensive **planning mechanisms** such as trajectory stitching and data-coherent path prediction. These additions enable our framework to perform effective hierarchical planning, which is not addressed by SAQ, thereby providing a more holistic solution for goal-conditioned offline RL.
>  6. **SkillDiffuser [6]**
>      - **Overview:** SkillDiffuser utilizes diffusion-based task execution with skill abstractions for hierarchical planning, primarily in online RL settings.
>      - **Comparison:** SkillDiffuser’s diffusion-based approach and skill abstractions differ significantly from our method. Our work introduces **explicit trajectory stitching** and **contrastive loss**, which are specifically designed to enhance planning robustness and data alignment in offline RL. These features allow our framework to better handle the unique challenges of offline environments, such as data scarcity and the need for reliable trajectory generation based on existing data.
>  7. **TAP [7] (Transformer-Based Planning) (already in the related work)**
>      - **Overview:** TAP integrates a VQ-VAE between the encoder and decoder of a transformer, applying the transformer to sequences of (state, action, reward) tuples and planning based on expected returns.
>      - **Comparison:** While TAP leverages transformers for planning, our approach utilizes a **sequence generator** tailored for offline RL that emphasizes **data-aligned path prediction**. Unlike TAP’s focus on expected returns, our method prioritizes planning paths that are **supported by the dataset**, ensuring higher success rates by avoiding low-data regions. Additionally, our incorporation of **contrastive loss** enhances the temporal coherence of the learned representations, further distinguishing our approach from TAP’s methodology.
>
> **In summary**, while prior research has laid the groundwork for discrete planning and hierarchical RL, our work extends these concepts by introducing novel mechanisms tailored to the goal-conditioned offline RL setting. Through the integration of contrastive loss, explicit trajectory stitching, and data-coherent planning, we provide a robust and effective framework that addresses the specific challenges inherent to offline environments. These contributions not only differentiate our approach from existing methods but also advance the field by offering practical solutions for complex, long-horizon navigation tasks in offline RL.
>
> We thank again **R-XZDy** for those literature suggestions and  hope this detailed comparison clarifies the novel and impactful aspects of our work, demonstrating its significant advancement over existing methodologies in the field of goal-conditioned offline reinforcement learning.
>
> [1] Mazzaglia, Pietro, Tim Verbelen, and Bart Dhoedt. "Choreographer: learning and adapting skills in imagination." ICLR, the 11th International Conference on Learning Representations. 2023.
>
> [2] Hamed, Hany, et al. "Dr. Strategy: Model-Based Generalist Agents with Strategic Dreaming." Forty-first International Conference on Machine Learning.
>
> [3] Lee, Seungjae, et al. "Cqm: Curriculum reinforcement learning with a quantized world model." Advances in Neural Information Processing Systems 36 (2023): 78824-78845.
>
> [4] Yuan, Haoqi, et al. "Pre-training goal-based models for sample-efficient reinforcement learning." The Twelfth International Conference on Learning Representations. 2024.
>
> [5] Luo, Jianlan, et al. "Action-quantized offline reinforcement learning for robotic skill learning." Conference on Robot Learning. PMLR, 2023.
>
> [6] Liang, Zhixuan, et al. "Skilldiffuser: Interpretable hierarchical planning via skill abstractions in diffusion-based task execution." Proceedings of the IEEE/CVF Conference on Computer Vision and Pattern Recognition. 2024.
>
> [7] Zhang, Tianjun, et al. "Efficient Planning in a Compact Latent Action Space." The Eleventh International Conference on Learning Representations

---

> > ### Comment · Reviewer_XZDy · 2024-12-03
> > **Official Comment by Reviewer XZDy for [6. About additional related works]**
> >
> > Thanks to the authors for providing this extensive comparison.

---

> ### Comment · Reviewer_XZDy · 2024-12-03
> **Official Comment by Reviewer XZDy for [2. About codebook size and continuous planning]**
>
> **About codebook size**
>
> The provided results look useful for understanding the required codebook size and how your method works with more codes. Thank you for providing this to answer my question.
>
>
> **About the continuous planning**
>
> I do not understand why you compared: BC+IQL w/ repr. vs. BC+IQL w/o repr. vs. QPHIL w/ aug. vs. QPHIL w/o aug. Are not the augmentation and representation different things? Should not you fix the augmentation and compare for example,
>
> (with augmentation) BC+IQL w/ repr. vs. BC+IQL w/o repr. vs. QPHIL w/ repr. vs. QPHIL w/o repr
>
> and maybe another table for
>
> (w/o augmentation)  BC+IQL w/ repr. vs. BC+IQL w/o repr. vs. QPHIL w/ repr. vs. QPHIL w/o repr?
>
> Is not the augmentation can be applied to BC+IQL as well?

---

> > ### Author Response · Authors · 2024-12-04
> >
> > Thank you for your answers. The data augmentation and the learning of representations are two variants of respectively QPHIL and BC+IQL. QPHIL's data augmentation corresponds to the stitching of token trajectories in the high level, while BC+IQL's learning of representations corresponds to the learning of representations from HIQL that permits general subgoal generation.
> >
> > **For BC+IQL's learning of representations:** Since BC+IQL is derived from HIQL, it employs two policies: a high-level policy, $\pi_{\theta_h}^h(s_{t+k}|s_t,g)$, which generates a subgoal $s_{t+k}$ based on the current state $s_t$ and the goal $g$, and a low-level policy, $\pi_{\theta_l}(a_t|s_t,s_{t+k})$, which determines the action $a_t$ required to reach the subgoal $s_{t+k}$ from $s_t$. However, generating subgoals directly in raw observation spaces can be challenging, especially in high-dimensional environments. To address this, [1] proposes learning a state representation $\phi(s)$, resulting in a high-level policy $\pi_{\theta_h}^h(\phi(s_{t+k})|s_t,g)$ and a low-level policy $\pi_{\theta_l}^l(a_t|s_t,\phi(s_{t+k}))$. This modification forms the “w/ repr.” variant, leveraging learned representations to simplify subgoal generation.
> >
> > **For QPHIL's data augmentation:** As QPHIL learns a high level policy that plans on discrete subgoals, its purpose is to generate sequences of discrete tokens. As such, we can augment the dataset by stitching tokenized trajectories.
> >
> > The explicit stitching data augmentation is harder for BC+IQL since subgoals are continuous which limits its applicability while representations for discrete subgoals are not necessary for the discrete tokens of QPHIL.
> >
> > [1] HIQL: Offline Goal-Conditioned RL with Latent States as Actions

---

> ### Comment · Reviewer_XZDy · 2024-12-03
> **Official Comment by Reviewer XZDy for [1. On the selection of evaluation benchmarks]**
>
> > It provides a first rigorous and diverse testing ground for QPHIL’s navigation scope, as well as being challenging even for state-of-the-art methods like HIQL.
>
> I agree with you to some extent, but I think you still need to show how this method can perform in different environments, not just navigation similar to HIQL, they showed a comparison in the Kitchen, Calvin and Procgen environments. I believe that such results will make your results much stronger and complete.

---

### Meta-Review · Area_Chair_QcfM · 2024-12-19

**Metareview:**

This paper introduces QPHIL, combining state space discretization with transformer-based planning for offline goal-conditioned RL. While reviewers acknowledged the method's effectiveness on long-horizon AntMaze environments and the thorough ablation studies, significant concerns were raised about its broader applicability. The empirical validation is restricted to navigation tasks with clear spatial structure, making it difficult to assess the method's generalizability to other domains with high-dimensional observations or different underlying structures. Although the authors provided additional experiments and analyses in their response, including comparisons with Option-Critic and studies of tokenization strategies, the fundamental limitation of domain specificity remains unaddressed. The method also shows limited advantages over existing approaches like HIQL on smaller-scale problems.

Given these substantial limitations in demonstrating broader impact and applicability, I recommend rejection.

**Additional Comments On Reviewer Discussion:**

While reviewers acknowledged the method's effectiveness on long-horizon AntMaze environments and the thorough ablation studies, significant concerns were raised about its broader applicability. The empirical validation is restricted to navigation tasks with clear spatial structure, making it difficult to assess the method's generalizability to other domains with high-dimensional observations or different underlying structures. Although the authors provided additional experiments and analyses in their response, including comparisons with Option-Critic and studies of tokenization strategies, the fundamental limitation of domain specificity remains unaddressed. The method also shows limited advantages over existing approaches like HIQL on smaller-scale problems.

---

### Decision · Program_Chairs · 2025-01-22

Reject